# ON CONVEX DECISION REGIONS IN DEEP NETWORK REPRESENTATIONS

**Lenka Tětková**
lenhy@dtu.dk

**Thea Brüsch**
theb@dtu.dk

**Teresa Karen Scheidt**
tksc@dtu.dk

**Fabian Martin Mager**
fmager@dtu.dk

**Rasmus Ørtoft Aagaard**

**Jonathan Foldager**
jonathan.foldager@gmail.com

**Tommy Sonne Alstrøm**
tsal@dtu.dk

**Lars Kai Hansen**
lkai@dtu.dk

Section for Cognitive Systems, DTU Compute,
Technical University of Denmark
2800 Kongens Lyngby, Denmark

## ABSTRACT

Current work on human-machine alignment aims at understanding machine-learned latent spaces and their correspondence to human representations. Gärdenfors' conceptual spaces is a prominent framework for understanding human representations. Convexity of object regions in conceptual spaces is argued to promote generalizability, few-shot learning, and interpersonal alignment. Based on these insights, we investigate the notion of convexity of concept regions in machine-learned latent spaces. We develop a set of tools for measuring convexity in sampled data and evaluate emergent convexity in layered representations of state-of-the-art deep networks. We show that convexity is robust to relevant latent space transformations and, hence, meaningful as a quality of machine-learned latent spaces. We find that approximate convexity is pervasive in neural representations in multiple application domains, including models of images, audio, human activity, text, and medical images. Generally, we observe that fine-tuning increases the convexity of label regions. We find evidence that pretraining convexity of class label regions predicts subsequent fine-tuning performance.

## 1 INTRODUCTION

Understanding the barriers to human-machine alignment is as important as ever (see, e.g., Bender et al. (2021); Mahowald et al. (2023)). Representational alignment is a first step towards a greater goal of understanding value alignment (Christian, 2020). For understanding of alignment, it is fundamental to establish a common language for the regularities observed in human and machine representations. Here, we motivate and introduce the concept of convexity of object regions in machine-learned latent spaces.

Representational spaces in the brain are described in several ways, for example, geometric psychological spaces informed by similarity judgements or, based in the neurosciences, representations derived from the measurement of neural activity (Balkenius & Gärdenfors, 2016; Tang et al., 2023). Conceptual spaces as proposed by Gärdenfors are a mature approach to the former, i.e., human-learned geometrical representations of semantic similarity (Gärdenfors, 2014). The geometrical approach is rooted in work Shepard (1987), which opens with the important observation: "Because any object or situation experienced by an individual is unlikely to recur in exactly the same form

and context, psychology's first general law should, I suggest, be a law of generalization". This leads Shepard to favor geometrical representations in which concepts are represented by extended regions rather than single points, to allow for robust generalization. This is the view that has been comprehensively expanded and quantified in Gärdenfors (2014). The cognitive science insights are complemented by extant work investigating alignment between learned representations in machine and human conceptual spaces (Chung & Abbott, 2021; Goldstein et al., 2022; Valeriani et al., 2023), and numerous specific properties of the latent geometrical structure have been studied, such as the emergence of semantic separability in machine latent representations (Mamou et al., 2020). New insights in the representational geometry are found using the intrinsic dimension measure (Valeriani et al., 2023). The relevant geometries are not necessarily flat Euclidean spaces but are often better described as general manifolds (Hénaff & Simoncelli, 2016; Arvanitidis et al., 2018). In fact, Hénaff et al. (2019) suggests that semantic separability emerges by flattening or straightening trajectories in latent spaces, such as was earlier proposed for machine representations (Brahma et al., 2015). Similar reasoning was crucial for early methodological developments like ISOMAP (Tenenbaum et al., 2000) and kernel methods (Mika et al., 1998).

## 1.1 CONVEXITY IN CONCEPTUAL SPACES

Based on Shephard's idea of objects as extended regions, Gärdenfors formulated the hypothesis that *natural* concepts form convex regions in human geometrical representations (Gärdenfors, 1990; 2014; Warglien & Gärdenfors, 2013; Douven et al., 2022). Strößner (2022) elaborated on the notion of natural concepts as a social construct: "[Natural concepts] are often found in the core lexicon of natural languages—meaning that many languages have words that (roughly) correspond to such concepts—and are acquired without much instruction during language acquisition." One way to interpret the naturalness notion is to link it to independent physical mechanisms with macroscopic effects, i.e., effects that will be visible to all, hence, likely to appear in joint vocabularies. Such independent mechanisms play a core role in causal modeling (Parascandolo et al., 2018). A more low-level interplay between human and machine conceptual representations was discussed in Bechberger & Kühnberger (2022) with a specific focus on grounding shape spaces. The work reports good correspondences between human shape representations as obtained by pairwise similarity judgments and machine representations of the shape obtained from supervised and unsupervised learning, however, without touching the question of the convexity of object regions in machines.

Convexity is closely related to generalization in cognitive systems (Gärdenfors, 2001; Gärdenfors et al., 2018). The defining property of convexity (see Definition 1) implies that categorization can be extended by interpolation. We also note that simple generalization based on closeness to prototypes leads to convex decision regions (Voronoi tesselation induces convex regions) (Gärdenfors & Williams, 2001). Interestingly, convexity is also claimed to support few-shot learning (Gärdenfors, 2001). When basic concepts are learned as convex regions, new labels can be formed by geometrically guided composition, leading to new convex regions (e.g., by conjunction) or by other inductions leading to sets of convex regions. Finally, it is observed that convexity supports communication and interaction and thus the negotiation of meaning between subjects and the emergence of socially universal concepts, i.e., natural concepts (Warglien & Gärdenfors, 2013).

The geometry-driven cognitive science insights motivate our investigation here: *Are generalizable, grounded decision regions implemented as convex regions in machine-learned representations?*

The convexity of decision regions in machine-learned representations has not been addressed before, and therefore, we first need to develop the required investigative tools. Our contributions include:

- Introduction of convexity as a new dimension in human-machine alignment.
- Recapitulation of the salient properties of convex sets in flat and curved spaces.
- Proofs that convexity is stable to relevant latent space transformations.
- Creating an efficient workflow to measure Euclidean and graph convexity of decision regions in latent spaces.
- Empirical evidence of pervasive convexity of decision regions in self-supervised models for images, audio, movement, text, and brain images.
- Empirical evidence that convexity of a class decision region in a pretrained model predicts labelling accuracy of that class following fine-tuning, see Figure 1.

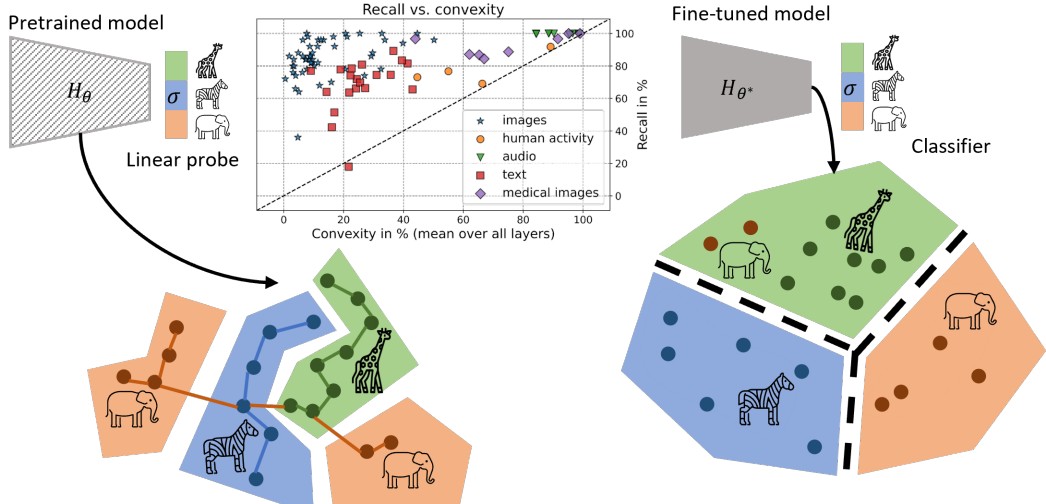

Figure 1: We measure the Euclidean and graph convexity of each class in the pretrained model and evaluate the recall of each class after fine-tuning. $H_\theta$ is the pretrained model, $H_{\theta^*}$ is the fine-tuned model, and $\sigma$'s are the classification heads trained to predict labels with fixed network representations ('pretrained') and during fine-tuning respectively. Fine-tuning involves all layers of the encoder. We present evidence (the inset and Figure 5) that higher convexity of downstream classes in the pretrained model is indicative of higher recall in the fine-tuned model. Hence, the convexity of a class decision region in the pretrained model is associated with subsequent learning of that class after fine-tuning.

## 1.2 PROPERTIES OF CONVEX SETS

Let us first formalize classical convexity in Euclidean spaces.

**Definition 1** (Euclidean convexity). *A subset $S \subset \mathbb{R}^D$ is convex iff $\forall \mathbf{x}, \mathbf{y} \in S \ \forall t \in [0, 1]$, $\mathbf{z}(t) = t\mathbf{x} + (1 - t)\mathbf{y}$ is also in S (Boyd & Vandenberghe, 2004).*

From the definition, it follows that the intersection of two convex sets is also a convex set. Hence, conceptual *conjunction* ('AND' operation) preserves convexity. *Disjunction* ('OR' operation), however, does not, since the union of convex sets is not necessarily convex (it is trivial to construct counter-examples) (Boyd & Vandenberghe, 2004). Euclidean convexity is conserved under affine transformations, hence convexity is robust to relevant latent space transformations in deep networks (see a more formal proof in Appendix A). Euclidean convexity is closely related to conjunctions of linear classifiers. In fact, a convex set can alternatively be defined as the intersection of linear half-spaces (possibly infinite), e.g., implemented by a set of linear decision functions resulting in a polyhedron (Boyd & Vandenberghe, 2004).

The relevant geometric structure of deep networks is not necessarily Euclidean, hence, we will also investigate convexity of decision regions in data manifolds. In a Riemannian manifold $M$ with metric tensor $g$, the length $L$ of a continuously differentiable curve $\mathbf{z} : [0, 1] \to M$ is defined by $L(\mathbf{z}) = \int_0^1 \sqrt{g_{\mathbf{z}(t)}(\dot{\mathbf{z}}(t), \dot{\mathbf{z}}(t))} dt$, where $\dot{\mathbf{z}}(t) := \frac{\partial}{\partial t}\mathbf{z}(t)$. A geodesic is then a curve connecting $\mathbf{z}(0) = \mathbf{x}$ and $\mathbf{z}(1) = \mathbf{y}$, minimizing this length, i.e. $\text{geodesic}(\mathbf{x}, \mathbf{y}) = \arg\min_{\mathbf{z}} L(\mathbf{z})$. While geodesics are unique for Euclidean spaces, they may not be unique in manifolds. We can now generalize to geodesic convexity in manifolds:

**Definition 2** (Geodesic convexity). *A region $S \in M$ is geodesic convex, iff $\forall \mathbf{x}, \mathbf{y} \in S$, there exists at least one geodesic $\mathbf{z}(t)$ connecting $\mathbf{x}$ and $\mathbf{y}$, that is entirely contained in S.*

When modeling latent spaces with sampled data, we must further transform the above definitions to data-driven estimators, such efforts are reported, e.g., in Hénaff & Simoncelli (2016); Arvanitidis et al. (2018). In this work, we choose a simple approach inspired by ISOMAP, hence based on graph convexity in data manifolds:

**Definition 3** (Graph convexity, see e.g., (Marc & Šubelj, 2018)). *Let $(V, E)$ be a graph and $A \subseteq V$. We say that $A$ is convex if for all pairs $x, y \in A$, there exists a shortest path $P = (x = v_0, v_1, v_2, \ldots, v_{n-1}, y = v_n)$ and $\forall i \in \{0, \ldots, n\} : v_i \in A$.*

For reasonably sampled data, we can form a graph based on Euclidean nearest neighbors (as in ISOMAP).

We note two important properties of this estimator, first, the graph-based approximate convexity measure is invariant to isometric transformation and uniform scaling, and second, the sample-based estimator of convexity is consistent. Both aspects are discussed further in Appendix A.

As we will measure convexity in labelled sub-graphs within larger graphs, Dijkstra's algorithm is preferred over Floyd–Warshall algorithm used in ISOMAP. Dijkstra's algorithm finds the shortest path from a given node to each of the other $N$ nodes in the graph with $E$ edges in $\mathcal{O}(N \log N + E)$ (Dijkstra, 1959; Fredman & Tarjan, 1987), while Floyd-Warshall efficiently finds the shortest distance between all vertices in the graph in $\mathcal{O}(N^3)$(Cormen et al., 2022; Floyd, 1962). As we have a sparse graph with $E \ll N^2$, Dijkstra's algorithm will be more efficient. With these approximations, we are in a position to create a graph-based workflow for quantifying convexity in Euclidean and manifold-based structures. Note, for sampled data, we expect a certain level of noise, hence, convexity will be graded.

### 1.2.1 CONSISTENCY OF GRAPH ESTIMATES OF GEODESICS

The consistency of sample/graph-based geodesic estimates has been discussed in connection with the introduction of ISOMAP (Bernstein et al., 2000) and more generally in Davis & Sethuraman (2019). Graph connectivity-based estimates of geodesics from sample data are implemented using two procedures: The neighborhood graph can be determined by a distance cutoff $\epsilon$, so that any points within Euclidean distance $\epsilon$ are considered neighbors, or by K-nearest neighbors (KNN) based on Euclidean distance. Consistency of these estimates, i.e., that the sample-based shortest paths converge to geodesics, is most straightforward to prove for the former approach (Bernstein et al., 2000; Davis & Sethuraman, 2019). The consistency proof for the $\epsilon$-based procedure is based on the smoothness of the metric, a uniformly bounded data distribution ($1/c < p(x) < c$) and scaling of the distance cutoff or $K$ so the connectivity (number of edges per node) increases for large samples (cutoff decays slowly $\epsilon \to 0$ as sample size $N \to \infty$) (Davis & Sethuraman, 2019).

In finite samples, a (too) large connectivity will bias the geodesics, while a (too) small connectivity can lead to disconnected graphs and noisy estimates. In pilot experiments, we tested the $\epsilon$ and KNN approaches for a range of $\epsilon$ and $K$ and found that the KNN approach produces more stable results. Moreover, the differences for various values of $K$ are negligible, so we are using KNN-based approach with $K = 10$ in the complete set of experiments. See Appendix C.6 for more details.

### 1.3 NEURAL NETWORKS AND CONVEXITY

Should we expect convexity of decision regions of neural network representations? Indeed, there are several mechanisms that could contribute to promoting convexity. First, the ubiquitous softmax is essentially a convexity-inducing device, hence, typical classification heads will induce convexity in their pre-synaptic activation layer. This is most easily seen by noting that softmax decision regions (maximum posterior decisions) are identical to the decision regions of a linear model, and linear models implement convex decision regions (see Appendix A). Secondly, several of our models are based on transformer architectures with attention heads. These heads contain softmax functions and are thus inducing convexity in their weighing of attention. Thirdly, typical individual artificial neurons, e.g., ReLUs, are latent half-space detectors and half-spaces are convex as noted above. Note that deep multi-layer perceptrons can approximate any non-convex decision region, including disconnected decision regions (Bishop et al., 1995). These findings suggest that convexity is possible in neural networks, even though it is not forced by design.

An extensive body of work concerns the geometry of input space representations in ReLU networks and has led to a detailed understanding of the role of so-called 'linear regions': In networks based on ReLU non-linearity, the network's output is a piece-wise linear function over convex input space polyhedra, formed by the intersection of neuron half-spaces (Montufar et al., 2014; Hanin

& Rolnick, 2019; Goujon et al., 2022; Fan et al., 2023). Interestingly, in typical networks (e.g., at initialization or after training), the linear regions are much less in number and simpler in structure compared to theoretical upper bounds (Hanin & Rolnick, 2019; Goujon et al., 2022; Fan et al., 2023). During training, the collection of convex linear regions is transformed and combined into decision regions through the later network layers. The resulting decision regions are, therefore, unions of convex sets and may in general be non-convex or non-connected, as noted in Bishop et al. (1995). Our two measures of convexity, Euclidean and graph-based, probe different mechanisms of generalization, the former is associated with generalization by linear interpolation, while the latter is associated with generalization by interpolation along the data manifolds. As both mechanisms may be relevant for explaining generalization in cognitive systems, we are interested in the abundance and potential role of both measures of convexity. Note that a region may be convex in terms of the Euclidean measure but not the graph-based (e.g. a decision region formed by disconnected but linearly separated subsets) and a region may be graph convex, but not Euclidean convex (e.g. a general shaped connected region in the data manifold). For visual examples see Figure 2.

## 2 METHODS

### 2.1 CONVEXITY MEASUREMENT WORKFLOWS

#### 2.1.1 GRAPH CONVEXITY

We are interested in measuring the approximate convexity of a decision region, here, a subset of nodes in a graph. A decision region for a specific class is a set of points that the model classifies as belonging to this class.

We first create a graph that contains all the data points of interest. The points are nodes in the graph and the Euclidean distances between the nodes are the weights of the edges. To handle manifold-based representation, for each node, we create an undirected edge only to the nearest neighbors ($K = 10$). This procedure creates a sparse undirected weighted graph with positive weights only.

We now sample $N_s$ pairs of points within the given predicted class label and compute the shortest path in the graph between the pairs using Dijkstra's algorithm (Dijkstra, 1959). For each path, we compute a score between 0 and 1. The path score is defined as the proportion of the number of nodes on the shortest path, without the endpoints, inside the decision region. If an edge directly connects the pair of nodes, the score is 1. If the points are not connected, the score is 0. We average the scores for all paths and all classes and get one number per layer. Error bars in the results show the standard error of the mean, where we set $n$ to the number of points in the given class. This is likely a conservative estimate since the mean is based on many more pairs than there are points.

The runtime complexity of this procedure is $\mathcal{O}\left(N^2(D + 1) + CN_s(N_s - 1)N\left(\log N + K\right)\right)$, where $N$ denotes the number of data points, $D$ is the dimensionality of the representations, and $C$ is the number of classes. See Appendix A.4 for details.

#### 2.1.2 EUCLIDEAN CONVEXITY

Euclidean convexity is also measured for each layer and class separately. To measure the convexity of class $c$ in layer $l$, we first extract hidden representations at $l$ of all points predicted to belong to class $c$. We sample $N_s$ pairs of points. For each pair, we compute $N_p = 10$ equidistant points on the segment connecting the representations of the two endpoints. We feed the interpolated points to the rest of the network (after layer $l$). The score for each pair is then the proportion of the interpolated points that are also predicted to belong to class $c$. Finally, to get the score of Euclidean convexity for the whole class, we average the scores over all the pairs of points.

The runtime complexity of this procedure is $\mathcal{O}\left(CN_sN_pD\right)$, where $C$ denotes the number of classes, and $D$ is the dimensionality of the representations. See Appendix A.4 for details.

#### 2.1.3 PROPERTIES OF THE CONVEXITY SCORES

To gain more intuition about the two types of convexity and their differences, present examples of various properties of the graph and Euclidean convexity score on synthetic data. Figure 2 shows how

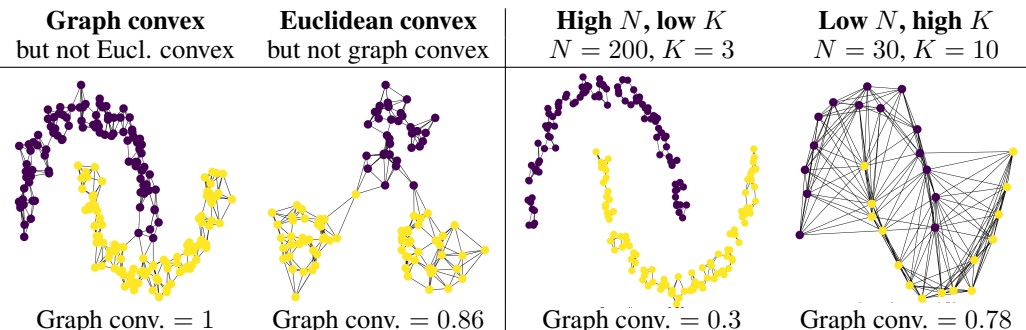

| **Graph convex** but not Eucl. convex | **Euclidean convex** but not graph convex | **High $N$, low $K$** $N = 200, K = 3$ | **Low $N$, high $K$** $N = 30, K = 10$ |
|---|---|---|---|
| Graph conv. = 1 | Graph conv. = 0.86 | Graph conv. = 0.3 | Graph conv. = 0.78 |

Figure 2: Properties of the graph convexity score: (left) Graph convexity can be more general than Euclidean convexity for a sufficiently large number of data points, $n$, and an appropriate number of nearest neighbors, $K$; A Euclidean convex set can have a graph convexity score lower than 1 in the case of isolated subgraphs of one class, connected only through the other class; (right) Low $n$ leads to edges to distant points that are then used in the shortest paths; Low $K$ can create a disconnected graph with low graph convexity score due to missing paths between the nodes from the same class.

different values of the sampled data and nearest neighbors influence the created graph and, therefore, also the graph convexity score.

Note that we could consider two types of class labels: data labels (true classes) and model labels determined by the predictions of a model (decision regions). Figure 24 in Appendix C.6 illustrates the difference between these two in the last layer of a model. From Theorem 1 in Appendix A, we have that the last layer is always Euclidean convex for model labels. In previous work (Anonymous, 2023), we considered data labels. In the present work, we focus on decision regions defined by model labels as they directly probe model representations and decision processes. Model labels are more 'expensive' to compute as propagation of many patterns through deep networks is required for Euclidean convexity.

For the pretrained models, we obtain the predicted labels by training a linear layer on top of the last hidden layer (with the rest of the model frozen). This procedure is similar to linear-classifier-based probes that are widely used in natural language processing to understand the presence of concepts in latent spaces (Belinkov et al., 2017; Hewitt & Liang, 2019). A prominent example of probe-based explanation in image classification is the TCAV scheme proposed by Kim et al. (Kim et al., 2018), in which auxiliary labelled data sets are used to identify concept directions with linear classifiers (concept class versus random images). In our approach, we use a multi-label "probe" of the last layer to identify the model labels but we use these labels to compute and compare convexity throughout the network.

The score depends on the number of classes, and also on the number of predicted data points per class. It follows that the exact numbers are not directly comparable across modalities. A scale for convexity can be set using a null hypothesis that there is no relation. Under this null, we can estimate convexity with randomized class assignments and get a baseline score (see Appendix C.7 for more details).

Measurements based on neighbors in high-dimensional data can be sensitive to the so-called hubness problem (Radovanović et al., 2010). We evaluate the hubness of the latent representations in terms of k-skewness and the Robinhood score. Results are deferred to Appendices C.1-C.5 since only mild hubness issues were detected for most domains. We decided to analyse convexity without adjustment for hubness, to avoid possible biases introduced by normalization schemes (Radovanović et al., 2010).

## 2.2 DOMAINS AND DATA

**Image domain**. We used ImageNet-1k images and class labels (Russakovsky et al., 2015; Deng et al., 2009) in our experiments. The validation set contains 1000 classes with 50 images per class. The network model is data2vec-base (Baevski et al., 2022). For details on architecture and training,

see Appendix B.1. We extracted the input embedding together with 12 layers of dimension 768 for geometric analysis.

**Human activity domain**. In the human activity domain, we applied our methods to the Capture24 dataset (Willetts et al., 2018). The dataset consists of tri-axial accelerometer data from 152 participants, recorded using a wrist-worn accelerometer in a free-living setting. In Walmsley et al. (2022), the dataset was annotated for four classes, namely; sleeping, sedentary behavior, light physical activity behaviors, and moderate-to-vigorous physical activity behaviors, which we use for fine-tuning. We formed the graph by sampling 5000 points (or the maximum number available) from each of the decision regions. We sampled 5000 paths from each label class during convexity analysis.

We used a pretrained human activity model from Yuan et al. (2022) to extract the latent representations. The model is pretrained in a self-supervised manner on a large unlabelled dataset from the UK Biobank. The model follows the architecture of ResNet-V2 with 1D convolutions and a total of 21 convolutional layers. The resulting feature vector after the final pretrained layer is of dimension 1024. For additional information on the network and the data see Appendix B.2.

**Audio domain**. In the audio domain, we used the wav2vec2.0 model (Baevski et al., 2020), pretrained on the Librispeech corpus consisting of 960h of unlabeled data (Panayotov et al., 2015). The model consists of a CNN-based feature encoder, a transformer-based context network, and a quantization module. We were especially interested in the latent space representation in the initial embedding layer and the 12 transformer layers. After each layer, we extracted the feature vector of dimension 768.

We fine-tuned the model to perform digit classification based on the AudioMNIST dataset (Becker et al., 2018), which consists of 30000 audio recordings of spoken digits (0-9) in English of 60 different speakers (with 50 repetitions per digit). For additional information, see Appendix B.3.

**Text domain**. Our NLP case study is on the base version of RoBERTa (Liu et al., 2019) which is pretrained to perform Masked Language Modelling (Devlin et al., 2018) in order to reconstruct masked pieces of text. The model consists of an embedding layer followed by 12 transformer encoder layers. Fine-tuning was done on the 20 newsgroups dataset (Lang, 1995) which consists of around 18,000 newsgroups posts, covering 20 different topics, which was split into train, validation and test sets. For further details see Appendix B.4.

**Medical imaging domain**. For the medical imaging domain, we used digital images of normal peripheral blood cells (Acevedo et al., 2020). The dataset contains 17,092 images of eight normal blood cell types: neutrophils, eosinophils, basophils, lymphocytes, monocytes, immature granulocytes (ig), erythroblasts and platelets. We fully pretrain a base-version of I-JEPA (Assran et al., 2023), a self-supervised transformer-based architecture with 12 transformer blocks, a patch size of 16, and a feature dimension of 768. We used 80% of the available data for a total of 150 epochs for pretraining. We evaluated the pretrained model by freezing its weights, averaging over the patch dimension of the last layer of the encoder, and applying a linear classification probe. Equivalently, during fine-tuning, we attached a linear classifier and retrained the whole model. We validated our model on 1709 hold-out samples. We achieved a pretrained accuracy of 85.3% and a fine-tuning accuracy of 93.5%. Embeddings were extracted from the first layer and every second transformer block, and averaged along the patch dimension.

## 3 RESULTS

To gain intuition on the effect of fine-tuning, we first inspect t-SNE plots for a subset of classes (Figure 3) of the image domain. From Theorem 1 in Appendix A, we learn that the last layer of both the pretrained and fine-tuned models is Euclidean convex. The t-SNE plot illustrate this quite clearly for the fine-tuned model while the picture is not quite as evident for the pretrained model. We also see that points with the same predictions get clustered within earlier layers. The situation is similar for other modalities (see Appendices C.1-C.5). The observed structure in these low-dimensional representations adds motivation to our investigation of the convexity of class labels using Euclidean and graph methods in the high-dimensional latent spaces.

We measure convexity for classes within the pretrained networks and after fine-tuning with class labels as targets. Figure 4 shows the results for all modalities. The convexity is pervasive both before

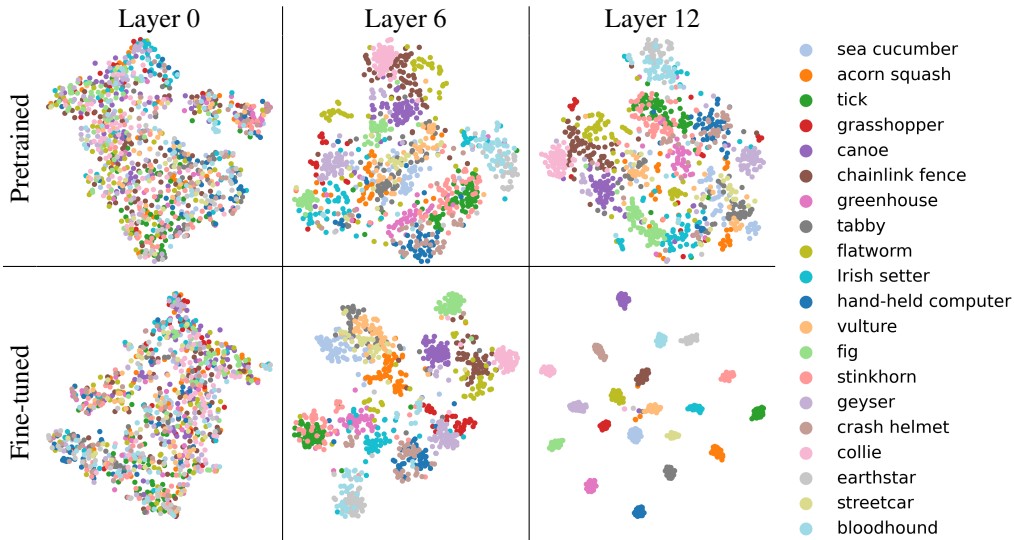

Figure 3: Images: t-SNE plots for a subset of predicted classes, both before and after fine-tuning.

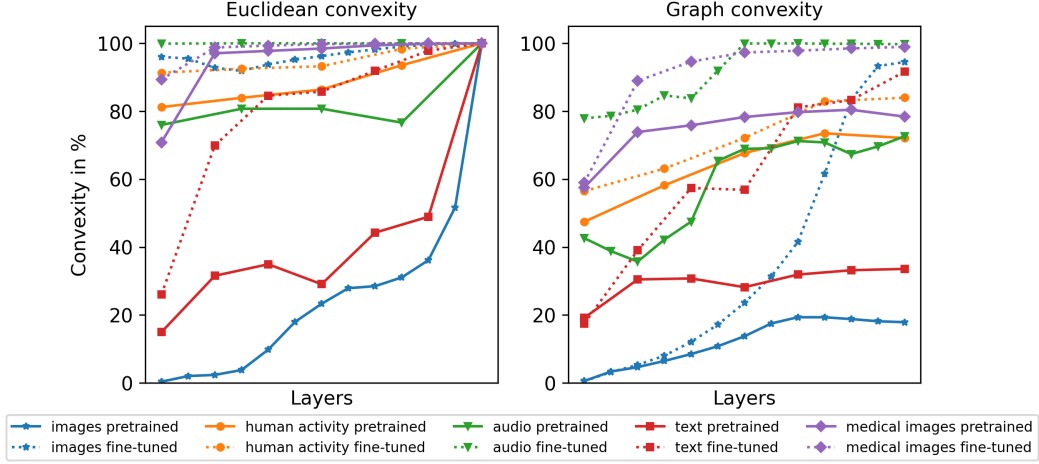

Figure 4: Euclidean and graph convexity scores for all modalities for decision regions of pretrained and fine-tuned networks. Decision regions are determined by model-predicted labels. Prediction in pretrained models is found using a softmax probe, training only the softmax linear layer. In fine-tuning, we train the whole network and softmax output. We find pervasive convexity in all networks and convexity further increases following fine-tuning. The number of layers differs across models but the most-left layer is the first layer that we observe and the right-most layer is the last layer in each model. Error bars are omitted in this plot for clarity (uncertainty estimates are given in figures of individual modalities in Appendices C.1-C.5). Note that the results are not directly comparable across modalities (see Section 2.1). In particular, we find less convexity in the image data containing a high number of classes (C=1000).

and after fine-tuning. An important reason why the magnitudes of the convexity scores differ among modalities is the different amounts of data and number of classes. However, the trend is the same for all modalities. Random labelling yields the graph convexity scores approximately $\frac{1}{C}$, where $C$ is the number of classes (more details on baseline scores in Appendix C.7). All the convexity scores are significantly higher than these baselines. Generally, class convexity is increased by fine-tuning. Note that the graph convexity in the last layer is (in some cases considerably) lower than $1 \sim 100\%$, even though this layer is always Euclidean convex.

For detailed results of the analysis of individual data modalities, see Appendices C.1-C.5.

To test the cognitive science-motivated hypothesis that convexity facilitates few-shot learning, we plot the post-finetuning accuracy (recall) per class versus pretraining convexity scores as shown in Figure 5 for graph-based convexity (top) and Euclidean convexity (bottom). There is a strong association between both types of convexity measured in the pretrained model and the accuracy of the given class in the fine-tuned model. Indeed, there is a significant correlation in both cases. The Pearson correlation between the two types of convexity scores is $0.57 \pm 0.04$ for both pretrained models and the fine-tuned ones. These results are quantifying the hypothesis that the two convexity scores are related but different. Detailed results can be found in Appendices C.1-C.5.

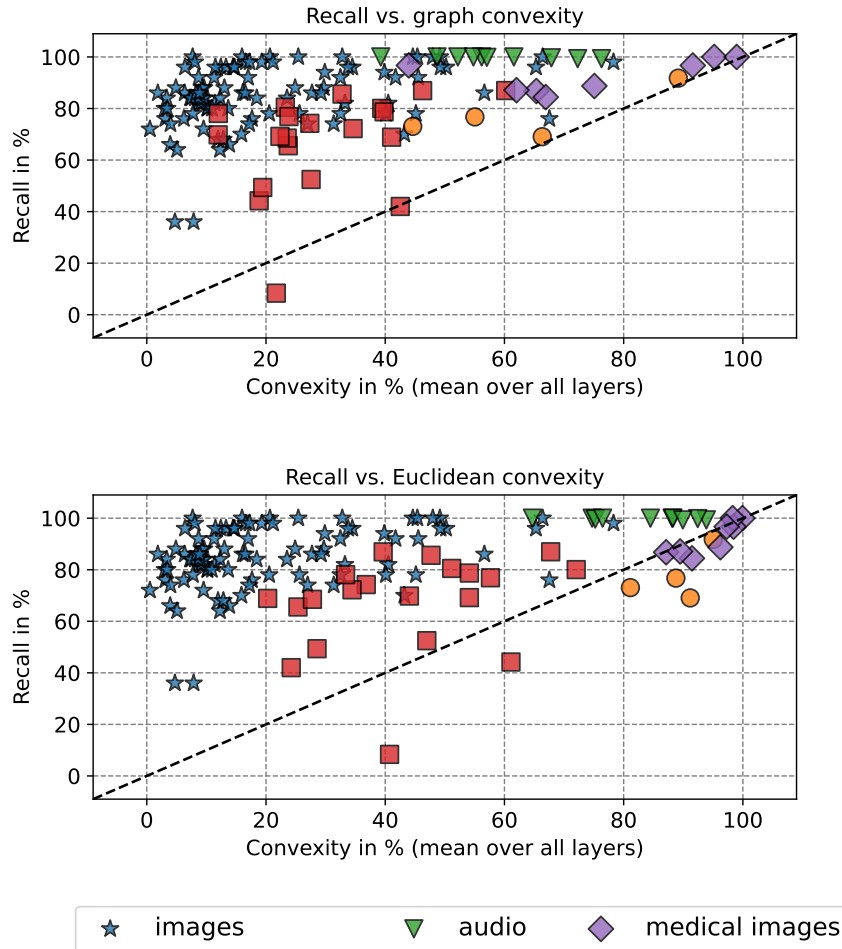

Figure 5: Graph convexity (top) and Euclidean convexity (bottom) of a subset of classes in the pretrained models vs. recall rate of these individual classes in the fine-tuned models for all data domains. The number of points for each domain is equal to the number of classes in this domain (except for images, where we take only a subset of classes for clarity – all image classes are shown in and Figure 9 and Figure 10 in Appendix C). The Pearson correlation coefficient is $0.22 \pm 0.06$ for graph convexity and $0.24 \pm 0.06$ for Euclidean convexity (the confidence intervals are computed using Fisher transformation).

## 4 CONCLUSION AND PERSPECTIVES

Understanding machine and human representational spaces is important for alignment and trust. Such investigations can be furthered by aligning the vocabularies of investigations into human and machine representations. Inspired by the conceptual space research of Gärdenfors and coworkers, we introduce the idea of convexity as a relevant dimension for machine-learned representations.

Convexity is closely related to generalization in cognitive systems, based on mechanisms such as generalization by interpolation or by proximity to prototype.

Machine representations are often found to be better described as curved spaces, hence, we recapitulated salient properties of convex sets in flat and curved spaces. In particular, we considered both conventional Euclidean and graph-based convexity. We found that convexity is stable to relevant latent space transformations for deep networks. We developed workflows for the estimation of approximate convexity based on Euclidean and graph methods for measuring convexity in latent spaces. We carried out extensive experiments in multiple domains including visual object recognition, human activity data, audio, text, and medical imaging. Our experiments included networks trained by self-supervised learning and next fine-tuned on domain-specific labels. We found evidence that both types of convexity are pervasive in both pretrained and fine-tuned models. On fine-tuning, we found that decision region convexity generally increases. Importantly, we find evidence that the higher convexity of a class decision region after pretraining is associated with the higher level of recognition of the given class after fine-tuning. This is in line with observations made in cognitive systems, that convexity supports few-shot learning.

## ACKNOWLEDGEMENTS

This work was supported by the DIREC Bridge project Deep Learning and Automation of Imaging-Based Quality of Seeds and Grains, Innovation Fund Denmark grant number 9142-00001B. This work was supported by the Pioneer Centre for AI, DNRF grant number P1 and the Novo Nordisk Foundation grant NNF22OC0076907 "Cognitive spaces - Next generation explainability". This work was supported by the Danish Data Science Academy, which is funded by the Novo Nordisk Foundation (NNF21SA0069429) and VILLUM FONDEN (40516). This work was partially supported by DeiC National HPC (g.a. DeiC-DTU-N5-20230028) and by the "Alignment of human and machine representations" project (g.a. DeiC-DTU-N5-20230033) and by the "self-supervised brain model" project (g.a. DeiC-DTU-S5-202300105). We acknowledge Danish e-infrastructure Cooperation (DeiC), Denmark, for awarding this project access to the LUMI supercomputer, owned by the EuroHPC Joint Undertaking, hosted by CSC (Finland) and the LUMI consortium through Danish e-infrastructure Cooperation (DeiC), Denmark, "Alignment of human and machine representations", DeiC-DTU-N5-20230028.

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

## APPENDICES

## A  THEORY

In this appendix, we present some mathematical background and intuition for the approximate convexity workflow.

### A.1  SOFTMAX INDUCES CONVEXITY

**Theorem 1.** *The preimage of each decision region under the last dense layer and softmax function is a convex set. More precisely: Let us denote the output of the last dense layer by*

$$a_k(\boldsymbol{z}) = \sum_{j=1}^{J} w_{k,j} z_j \qquad (1)$$

*for $\boldsymbol{z} \in \mathbb{R}^n$ and the probabilities*

$$p_k(\boldsymbol{z}) = \frac{\exp a_k(\boldsymbol{z})}{\sum_{k'=1}^{K} \exp a_{k'}(\boldsymbol{z})}. \qquad (2)$$

*Denote*

$$C_k \subseteq \mathbb{R}^n = \{\boldsymbol{x} \in \mathbb{R}^n : p_k(\boldsymbol{x}) > p_j(\boldsymbol{x}) \ \forall j \in \{1,\dots,K\}, j \neq k\}. \qquad (3)$$

*Then $C_k$ is a convex set for any $k \in \{1,\dots,K\}$ (Bishop & Nasrabadi, 2006).*

*Proof.* Let us define regions

$$\boldsymbol{x} \in \mathcal{R}_k \iff (a_k(\boldsymbol{x}) > a_j(\boldsymbol{x}) \ \forall j \neq k). \qquad (4)$$

Because $\sum_{k'=1}^{K} \exp a_{k'} > 0$ and by the monotonicity of the exponential function, it holds that

$$k_{opt}(\boldsymbol{x}) := \arg\max_k p_k(\boldsymbol{x}) = \arg\max_k \frac{\exp a_k(\boldsymbol{x})}{\sum_{k'=1}^{K} \exp a_{k'}(\boldsymbol{x})} = \arg\max_k a_k(\boldsymbol{x}), \qquad (5)$$

Therefore,

$$\mathcal{R}_k = \mathcal{C}_k \ \ \forall k \in \{1,\dots K\} \qquad (6)$$

and we can from now on work with $\mathcal{R}_k$.

Following the definitions from Bishop & Nasrabadi (2006), pages 182-184:
Let $\boldsymbol{x}_A, \boldsymbol{x}_B \in \mathcal{R}_k$ and define any point $\hat{\boldsymbol{x}}$ that lies on the line connecting the two:

$$\hat{\boldsymbol{x}} = \lambda \boldsymbol{x}_A + (1 - \lambda)\boldsymbol{x}_B, \qquad (7)$$

where $0 \leq \lambda \leq 1$. Cf. equation 1, the discriminant functions, $a_k$, are linear and it follows that:

$$a_k(\hat{\boldsymbol{x}}) = \lambda a_k(\boldsymbol{x}_A) + (1 - \lambda)a_k(\boldsymbol{x}_B) \tag{8}$$

for all $k$. From the definition of $\mathcal{R}_k$, we know that $a_k(\boldsymbol{x}_A) > a_j(\boldsymbol{x}_A)$ and $a_k(\boldsymbol{x}_B) > a_j(\boldsymbol{x}_B)$ for all $j \neq k$. Therefore, $a_k(\hat{\boldsymbol{x}}) > a_j(\hat{\boldsymbol{x}})$ for all $j \neq k$ and $\hat{\boldsymbol{x}}$ belongs to $\mathcal{R}_k$. Hence, $\mathcal{R}_k = \mathcal{C}_k$ is convex. □

## A.2 EUCLIDEAN CONVEXITY IS INVARIANT TO AFFINE TRANSFORMATIONS

**Theorem 2.** *Let $f : \mathbb{R}^n \to \mathbb{R}^n$ be an affine transformation (i.e., $f(\boldsymbol{x}) = \boldsymbol{A}\boldsymbol{x} + \boldsymbol{b}$, where $b \in \mathbb{R}^n$ and $\boldsymbol{A} : \mathbb{R}^n \to \mathbb{R}^n$ is an invertible linear transformation). Let $X \subset \mathbb{R}^n$ be a convex set. Then $f(X)$ is convex (Boyd & Vandenberghe, 2010).*

*Proof.* Let $\boldsymbol{x}, \boldsymbol{y} \in f(X)$ and $\lambda \in [0, 1]$. There exist $\overline{\boldsymbol{x}}, \overline{\boldsymbol{y}} \in X$ such that $f(\overline{\boldsymbol{x}}) = \boldsymbol{x}$ and $f(\overline{\boldsymbol{y}}) = \boldsymbol{y}$. Since $X$ is convex, we have $\lambda\overline{\boldsymbol{x}} + (1 - \lambda)\overline{\boldsymbol{y}} \in X$. Therefore,

$$f\left(\lambda\overline{\boldsymbol{x}} + (1 - \lambda)\overline{\boldsymbol{y}}\right) \in f(X). \tag{9}$$

Moreover, because of the linearity of $A$,

$$f\left(\lambda\overline{\boldsymbol{x}} + (1 - \lambda)\overline{\boldsymbol{y}}\right) = \boldsymbol{A}\left(\lambda\overline{\boldsymbol{x}} + (1 - \lambda)\overline{\boldsymbol{y}}\right) + \boldsymbol{b} = \lambda\boldsymbol{A}\overline{\boldsymbol{x}} + (1 - \lambda)\boldsymbol{A}\overline{\boldsymbol{y}} + \boldsymbol{b}. \tag{10}$$

Finally, we have

$$\lambda\boldsymbol{A}\overline{\boldsymbol{x}} + (1 - \lambda)\boldsymbol{A}\overline{\boldsymbol{y}} + \boldsymbol{b} = \lambda\left(\boldsymbol{A}\overline{\boldsymbol{x}} + \boldsymbol{b}\right) + (1 - \lambda)\left(\boldsymbol{A}\overline{\boldsymbol{y}} + \boldsymbol{b}\right) \tag{11}$$
$$= \lambda f(\overline{\boldsymbol{x}}) + (1 - \lambda)f(\overline{\boldsymbol{y}}) \tag{12}$$
$$= \lambda\boldsymbol{x} + (1 - \lambda)\boldsymbol{y}. \tag{13}$$

Hence,

$$f\left(\lambda\overline{\boldsymbol{x}} + (1 - \lambda)\overline{\boldsymbol{y}}\right) = \lambda\boldsymbol{x} + (1 - \lambda)\boldsymbol{y} \tag{14}$$

and $\lambda\boldsymbol{x} + (1 - \lambda)\boldsymbol{y} \in f(X)$. Therefore, $f(X)$ is convex. □

## A.3 GRAPH CONVEXITY IS INVARIANT TO ISOMETRY AND UNIFORM SCALING

**Theorem 3.** *Graph convexity is invariant to isometry and uniform scaling.*

*Proof.* Isometry (with respect to Euclidean distance) is a map $I : \mathbb{R}^n \to \mathbb{R}^n$ such that

$$\forall \boldsymbol{x}, \boldsymbol{y} \in \mathbb{R}^n : \|\boldsymbol{x} - \boldsymbol{y}\|_2 = \|I(\boldsymbol{x}) - I(\boldsymbol{y})\|_2. \tag{15}$$

Since isometry preserves distances, it induces the same distance graph. Therefore, graph convexity is invariant to isometries.

Universal scaling is a map $S : \mathbb{R}^n \to \mathbb{R}^n$ such that

$$\exists C \in \mathbb{R} \ \forall \boldsymbol{x} \in \mathbb{R}^n : \ S(\boldsymbol{x}) = C\boldsymbol{x}. \tag{16}$$

It holds that

$$\|S(\boldsymbol{x}) - S(\boldsymbol{y})\|_2 = \|C\boldsymbol{x} - C\boldsymbol{y}\|_2 = |C| \cdot \|\boldsymbol{x} - \boldsymbol{y}\|_2. \tag{17}$$

All the distances are scaled by $|C|$. It follows that the chosen nearest neighbours are the same. We show by contradiction that all the shortest paths are also the same. We denote $G$ the graph with the original edges and $G_S$ the graph with the scaled edges.

Given $\boldsymbol{x}, \boldsymbol{y} \in \mathbb{R}^n$, suppose that there exists a shortest path:

$$P = (\boldsymbol{p_0} = \boldsymbol{x}, \boldsymbol{p_1}, \ldots, \boldsymbol{p_j} = \boldsymbol{y}) \tag{18}$$

from $\boldsymbol{x}$ to $\boldsymbol{y}$, and a different path:

$$Q = (\boldsymbol{q_0} = S(\boldsymbol{x}), \boldsymbol{q_1}, \ldots, \boldsymbol{q_k} = S(\boldsymbol{y})) \tag{19}$$

from $S(\boldsymbol{x})$ to $S(\boldsymbol{y})$ such that

$$L(Q) < L(S(P)), \tag{20}$$

where
$$S(P) = (S(\boldsymbol{p_0}), S(\boldsymbol{p_1}), \ldots, S(\boldsymbol{p_j})) \tag{21}$$

and $L(\cdot)$ is an operator measuring the length of a path. From equation 20, it follows that $C \neq 0$. Because all the edges are scaled by a factor $|C|$, it holds that $L(S(P)) = |C|L(P)$. Moreover, for every pair of nodes $\boldsymbol{v_1}$, $\boldsymbol{v_2}$, there exists an edge in $G$ from $\boldsymbol{v_1}$ to $\boldsymbol{v_2}$ if and only if there exists an edge in $G_S$ from $S(\boldsymbol{v_1})$ to $S(\boldsymbol{v_2})$. Hence, there exists a path $R$ from $\boldsymbol{x}$ to $\boldsymbol{y}$ such that $Q = S(R)$. It follows that

$$|C|L(R) = L(S(R)) < L(S(P)) = |C|L(P). \tag{22}$$

Therefore, $L(R) < L(P)$ and that contradicts the assumption that $P$ is the shortest path from $\boldsymbol{x}$ to $\boldsymbol{y}$. $\qquad \square$

## A.4 RUNTIME COMPLEXITY

We have a dataset $\mathcal{D}$ with $C$ classes each with $N_c$ data points, $N$ total data points, and each data point has a representation in $\boldsymbol{z} \in \mathbb{R}^D$.

**Graph Convexity**: Computing the graph convexity score requires three steps:

1. Compute the nearest neighbor matrix. This cost $1/2N^2 D$.
2. Create the adjacency matrix. This cost $1/2N^2$. The adjacency matrix will at most have $E = NK$ edges, where $K$ is the $K$ chosen for $K$-nearest neighbors
3. Compute the convexity score for each pair of data points for each class or concept using Dijkstra's algorithm. This cost $1/2CN_c(N_c - 1)(N \log N + E)$.

The total computational cost is then

$$\text{cost} = 1/2N^2 D + 1/2N^2 + 1/2CN_c(N_c - 1)\left(N \log N + NK\right)$$

This simplifies to

$$\text{cost} = \mathcal{O}\left(N^2(D+1) + CN_c(N_c - 1)N\left(\log N + K\right)\right)$$

Since $CN_c = N$, then if $(N_c - 1)(\log N + K) \gg (D+1)$, the dominating term will be the latter. This can be seen by rewriting:

$$\text{cost} = \mathcal{O}\left(N^2(D+1) + N^2(N_c - 1)\left(\log N + K\right)\right)$$

Instead of computing the convexity score for a graph exact, we can estimate it using $N_s$ samples per class/concept, we get

$$\text{cost} = \mathcal{O}\left(N^2(D+1) + CN_s(N_s - 1)N\left(\log N + K\right)\right)$$

**Euclidean Convexity**: Computing the Euclidean convexity score for each class/concepts entails the following operations

1. Sample $N_p$ point pairs, denoting their representations $\boldsymbol{z_1}$ and $\boldsymbol{z_2}$. This operation scales with $N_p$.
2. Per point pair, generate $N_s$ representations equidistant on a line between $\boldsymbol{z_1}$ and $\boldsymbol{z_2}$. This scales with $N_s$.
3. For each generated representation, compute the classification. This scales with $D$.

Since all the costs are scaling linearly, the total cost of computing the Euclidean convexity score is

$$\text{cost} = \mathcal{O}\left(CN_s N_p D\right)$$

## B DATA SETS

### B.1 IMAGE DOMAIN

We used ImageNet-1k (Russakovsky et al., 2015; Deng et al., 2009) in our experiments. The validation set contains 50,000 images in 1000 classes, i.e. 50 images per class.

We used data2vec-base (Baevski et al., 2022) architecture. It consists of 12 Transformer layers (Vaswani et al., 2017) and was trained to produce the same representations for an input image and its masked version. It was pretrained on an unlabelled version of ImageNet-1k. For fine-tuning, a linear layer was added on top of the mean-pooled output of the last layer. Both the pretrained model[1] and the model fine-tuned[2] on ImageNet-1k (Russakovsky et al., 2015; Deng et al., 2009) were obtained from Hugging Face (Wolf et al., 2020). Details on pretraining and fine-tuning can be found in the original paper (Baevski et al., 2022). The final accuracy of the fine-tuned model on the validation set is 83.594%.

We extracted the input embedding together with 12 layers. For each layer, we averaged the hidden states across patches to get 768-dimensional feature vectors.

To get the predictions for the pretrained model, we trained a linear layer with a learning rate of 0.1, a polynomial scheduler, and a weight decay of 0.0001 for 1000 epochs with the whole training set of ImageNet-1k. The final accuracy on the validation set is 46.766%.

## B.2 HUMAN ACTIVITY DOMAIN

In the human activity domain, we used the pretrained model from Yuan et al. (2022) to extract the latent representations. The model is pretrained on a large unlabelled dataset from the UK Biobank, which contains 700,000 person-days of free-living tri-axial accelerometer data. The pretraining procedure is a multi-task self-supervised learning schedule, where the model is trained to predict whether a number of augmentations have been applied or not. The model follows the architecture of ResNet-V2 with 1D convolutions and a total of 21 convolutional layers. The resulting feature vector after the final pretrained layer is of dimension 1024.

The model is divided into 5 modules, 4 modules consisting of 2 ResNet blocks each and a final convolutional layer mapping the data to 1024 channels. We extracted the latent representations after each of the 5 modules. We added a single linear layer with softmax activation to obtain the decision regions for both the pretrained and the fine-tuned model. We flattened all the latent representations during the analysis.

For testing the methods, we used the Capture-24 dataset (Willetts et al., 2018). The Capture-24 dataset contains free-living wrist-worn activity tracking data from 152 participants. The participants were tracked for 24 hours and the data was subsequently humanly labelled into 213 categories based on a wearable camera also worn by the participants. Each of the 213 categories is associated with a metabolic equivalent of task (MET) score, which is a number describing the energy expenditure of each task. In Walmsley et al. (2022), the original 213 labels were divided into 4 coarse labels, namely; sleeping, sedentary behaviour, light physical activity behaviours and moderate-to-vigorous physical activity behaviours based on the MET scores. These 4 labels are used as classes when fine-tuning the model.

During fine-tuning, we randomly selected 30 subjects to hold out for testing. We optimized the entire network (encoder and classifier) jointly with a learning rate of $10^{-4}$. Following the authors in Yuan et al. (2022), we selected a small validation set and used early-stopping with a patience of 5 epochs. The same procedure was used to obtain the decision regions for the pretrained model, however, the weights of the encoder were frozen.

We achieved a balanced accuracy score of 77.0% when fine-tuning the entire model and 72.2% when optimizing only the last linear layer.

For each decision region, we sampled 5000 points (or the maximum number available). We then analyzed the convexity of each decision region after each of the 5 modules in the network.

## B.3 AUDIO DOMAIN

In the audio domain, we used the pretrained wav2vec2.0 model (Baevski et al., 2020), which is trained on the Librispeech corpus (960h of unlabeled data) (Panayotov et al., 2015). During pertaining, the model learns meaningful latent audio/speech representations, the exact training objectives can be found in Baevski et al. (2020). The model consists of a CNN-based feature encoder, a

---

[1]https://huggingface.co/facebook/data2vec-vision-base
[2]https://huggingface.co/facebook/data2vec-vision-base-ft1k

transformer-based context network and a quantization module. We were especially interested in the latent space representation in the 12 transformer layers. After each transformer layer, we extracted the feature vector of dimension 768.

We fine-tuned the model to perform digit classification based on the AudioMNIST dataset (Becker et al., 2018), which consists of 30000 audio recordings of spoken digits (0-9) in English of 60 different speakers (with 50 repetitions per digit). For fine-tuning, an average pooling layer and a linear layer were added to the network, to perform a classification task, the fine-tuning procedure was based on a tutorial[3]. The network was fine-tuned on 80% of the people while the remaining 20% were withheld for testing (resulting in 6000 audio files). Two different fine-tunings were performed. First, only the final linear layer was added and trained with a frozen model. This was done with a learning-rate of $1e^{-3}$ and early stopping. The final model reached an accuracy of 64%. In the second fine-tuning, only the initial CNN layers were frozen, while the transformer layers and the added linear layer were fine-tuned. The model was fine-tuned with early stopping for 1000 steps (batch-size 16), and the learning rate was set to $1e^{-4}$. No hyperparameter search was performed as the first fine-tuning already led to a very high accuracy of 99.89%. The latent representations of the test set were extracted before and after fine-tuning for both scenarios.

## B.4 TEXT DOMAIN

For text, we used the base version of RoBERTa[4] (Liu et al., 2019) which is pretrained to perform Masked Language Modelling (Devlin et al., 2018) in order to reconstruct masked pieces of text. The model consists of an embedding layer followed by 12 transformer encoder layers and a classification head. The pretraining of RoBERTa is performed on 160GB of uncompressed English-language text in order to learn latent representations of text which are expressed as 768-dimensional vectors. One notable difference that we make is that the classification head, which in a standard configuration of RoBERTa is a fully-connected dense layer followed by dropout and then lastly a linear projection head to the number of classes, we simply replace all of this with a single linear layer, projecting the hidden last hidden states into the classes.

Fine-tuning was done on the 20 newsgroups dataset (Lang, 1995) which consists of around 18,000 newgroups posts, covering 20 different topics. Recommended pre-processing steps were performed to remove headers, signature blocks, and quotations from each new article, as the model would otherwise be likely to overfit to features generated from those fragments. The data was split into a training, validation and test set with 10,000, 1,000 and 7,000 posts, respectively. The data and exact splits that were used are available on HuggingFace[5] The validation set was used to perform early stopping during training.

We found that the model stopped after 3 epochs and reached an accuracy of 68.9%. This was with a learning rate of $10^{-4}$ and weight-decay of $10^{-2}$. For the pretrained model, we froze everything up until the final linear projection and only trained the last linear layer. After training for 8 epochs, as dictated by early stopping, an accuracy of 60% was reached through this approach. The learning rate was increased to $10^{-2}$ when training only the linear projection head.

## B.5 MEDICAL IMAGING DOMAIN

We investigated digital images of normal peripheral blood cells (Acevedo et al., 2020). The dataset contains 17,092 images of eight normal blood cell types: neutrophils, eosinophils, basophils, lymphocytes, monocytes, immature granulocytes (ig), erythroblasts and platelets. Images were obtained with CellaVision DM96 in RGB colour space. The image size is 360 × 363 pixels. Images were labelled by clinical pathologists at the Hospital Clinic of Barcelona.

We fully pretrained the base version of I-JEPA (Assran et al., 2023) with a patch size of 16, 12 transformer blocks with a feature dimension 768. I-JEPA is a masked image model, where the pretext task is to predict embeddings of hold-out image context from the embeddings of the encoder. The model was trained using a smooth L1 loss with default settings in PyTorch. The targets were

---

[3]https://colab.research.google.com/github/m3hrdadfi/soxan/blob/main/notebooks/Eating_Sound_Collection_using_Wav2Vec2.ipynb
[4]https://huggingface.co/roberta-base
[5]https://huggingface.co/datasets/rasgaard/20_newsgroups

80% of the available data for a total of 150 epochs. Hyperparameters were kept consistent with the original work of Assran et al. (2023). We chose a learning rate of 0.001 with cosine decay and a linear warmup of 20 epochs. We used the AdamW optimizer.

After pretraining, we froze the model weights of the encoder and trained a linear layer for 50 epochs with the same optimizer and a learning rate of 0.005. The pretrained model with a linear layer achieves an accuracy of 85.3% on the validation set of 1709 subjects. Equivalently, during fine-tuning, we attached a linear classifier and retrained the whole model. The fine-tuned model reaches an accuracy of 93.5%.

We used the 1709 hold-out samples for the convexity analysis. To extract features, we averaged over the patch dimension of the embeddings, resulting in a 768-dimensional vector per subject and layer. We extracted these vectors after the convolutional patch embedding layer, after transformer blocks 2, 4, 6, 8, 10, and after normalisation of the 12th transformer block, resulting in 7 feature vectors per subject. We sampled 5000 paths for all convexity analyses.

## C DETAILED RESULTS

Figure 6 shows hubness metrics (k-skewness and Robinhood score) for all models and domains. It follows that hubness is not a problem for any of the domains.

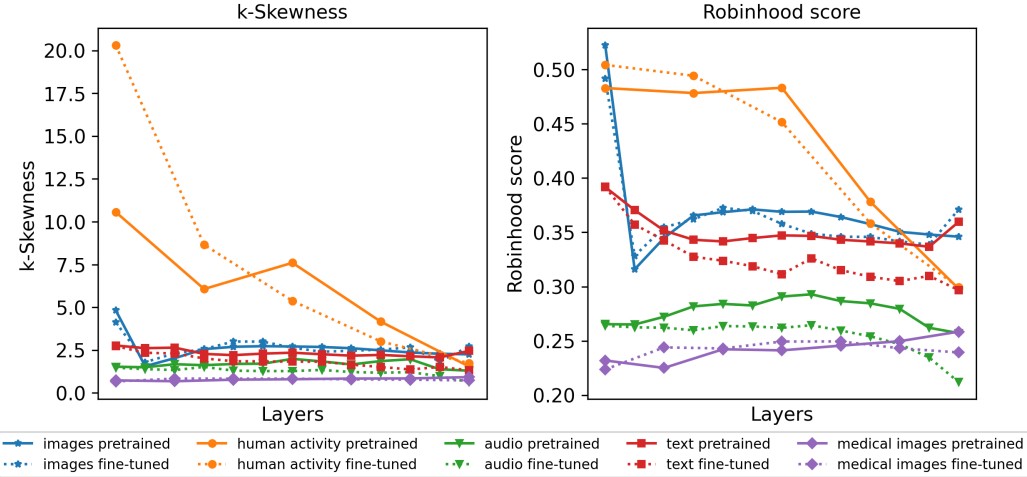

Figure 6: Hubness evaluation for all modalities. No measures to correct for hubness were taken, as the numbers for k-skewness and Robinhood score are generally low.

### C.1 IMAGE DOMAIN

In Figure 3, we show t-SNE plots for a subset of classes of the image domain with labels predicted by the models.

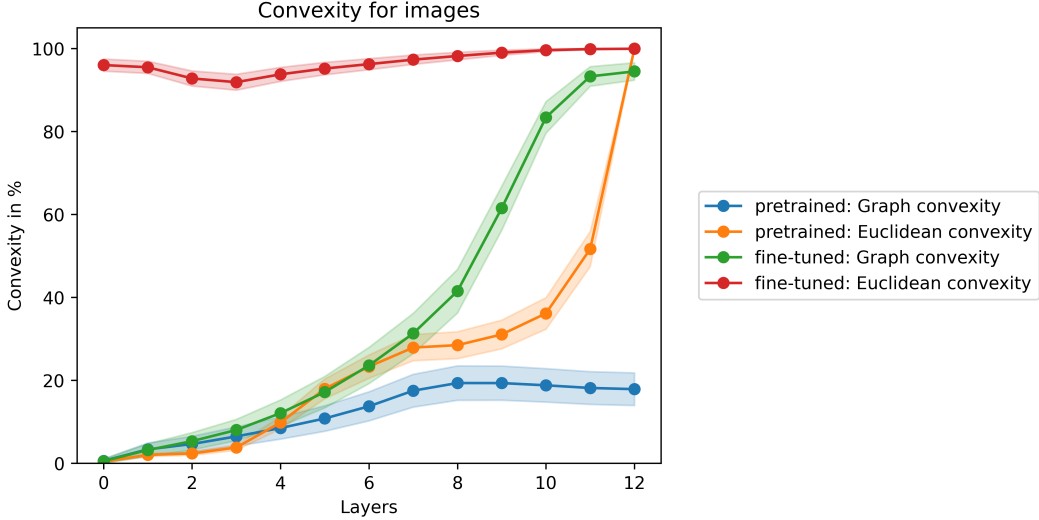

Figure 7: Images: Euclidean and graph convexity in the pretrained and fine-tuned model. Error bars show the standard error of the mean, where we set $n$ to the number of data points.

Figure 7 shows graph and Euclidean convexity results for both models. The relation between convexity in the pretrained model and accuracy per class in the fine-tuned model is depicted in Figure 9 for graph-based convexity and in Figure 10 for Euclidean convexity. Both convexity scores per class are plotted against each other in Figure 8.

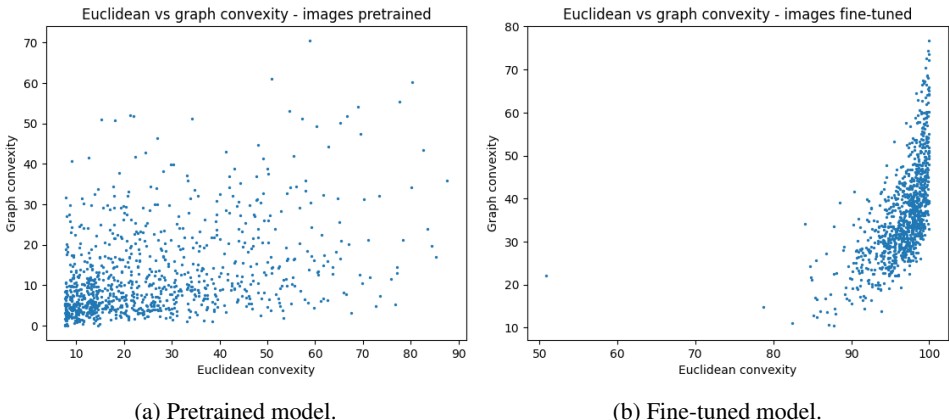

(a) Pretrained model.

(b) Fine-tuned model.

Figure 8: Images: Scatter plots of graph vs Euclidean convexity for each class in the pretrained and fine-tuned model.

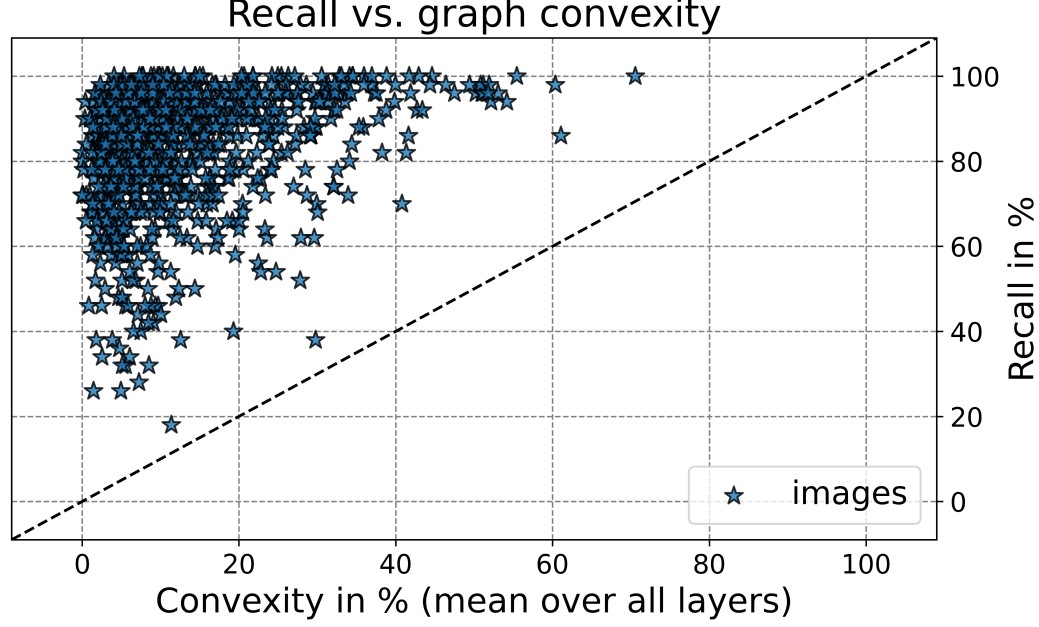

Figure 9: Graph convexity of all image classes in the pretrained models vs. recall of these classes in the fine-tuned models.

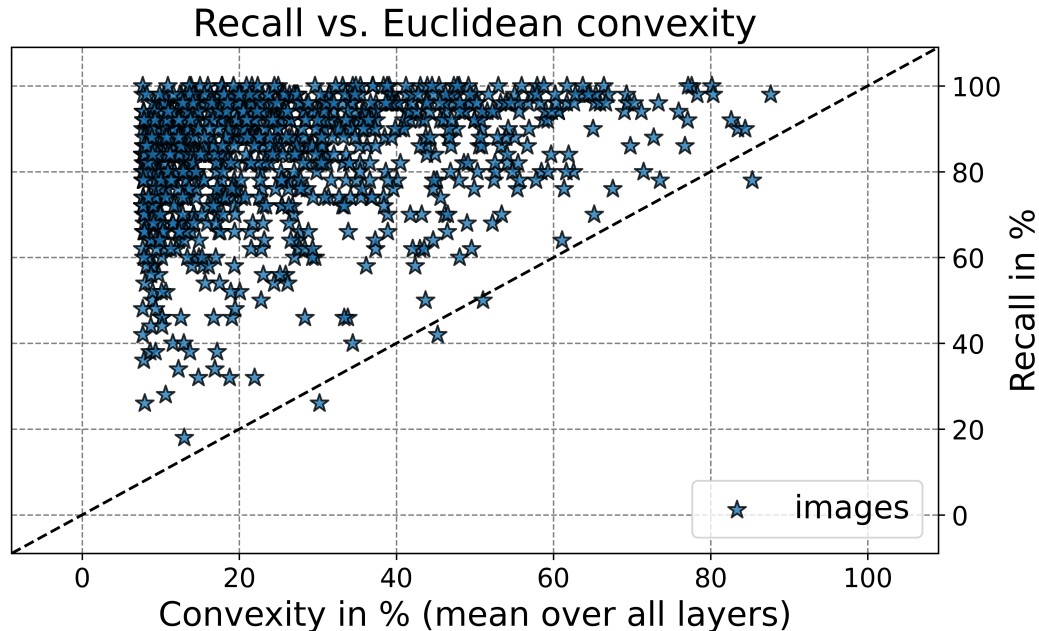

Figure 10: Euclidean convexity of all image classes in the pretrained models vs. recall of these classes in the fine-tuned models.

## C.2 HUMAN ACTIVITY DOMAIN

Figure 11 shows the t-SNE plots from the human activity domain with samples coloured by their predicted labels. The figure shows the tSNE plots of the representations after the first, third, and last (fifth) layers for both the pretrained and the fine-tuned model. For both models, it is clear that the decision regions change throughout the network. In the first layer, the decision regions are split into multiple different subregions. In the final layers, the regions appear more grouped, as would be expected. This is even more obvious in the fine-tuned model, where the decision regions also seem to have moved further apart. In the final layer, the decision regions are linearly separated in the true representation space. However, when projected into two dimensions by the t-SNE algorithm, it is clear that other structures in the data are also dominating the clustering.

Figure 12 shows the convexity analysis for both the pretrained and fine-tuned networks in the human activity domain. In both models, we notice a clear pattern of increasing graph and Euclidean convexity between the first and the last layer. The Euclidean convexity for the pretrained model starts at $\approx 80\%$ and ends, as expected, at $100\%$. For the fine-tuned model, the Euclidean convexity is even higher in the first layer and also increases to the expected $100\%$. Looking at the graph convexity scores, it is clear that these are lower compared to the Euclidean scores, and neither the pretrained nor the fine-tuned model have $100\%$ graph convex decision regions in the last layer. It is difficult to give the exact answer as to why, but a plausible suggestion could be that the decision regions contain multiple disconnected subregions. A synthetic example of such a case can be seen in Figure 2. This hypothesis could also be backed by the t-SNE plots, which indicate dominant substructures in the data that cause disconnectedness of data within decision regions. In general, these results indicate that the graph convexity score is able to capture other substructures in the data than what can be discovered from Euclidean convexity.

Figure 13 shows the relation between Euclidean and graph convexity per class.

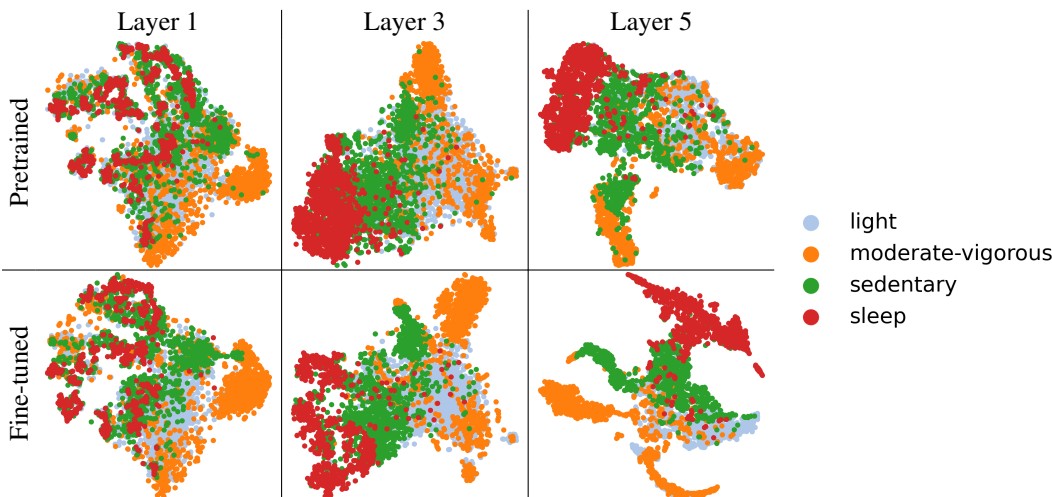

Figure 11: Human activity: t-SNE plots for a subset of samples from the predicted classes, both before and after fine-tuning.

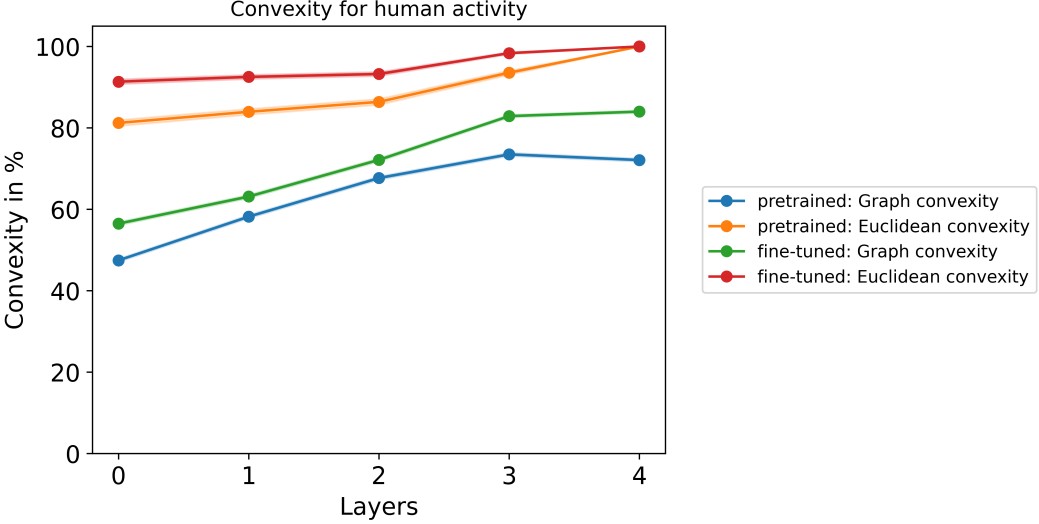

Figure 12: Human activity: Euclidean and graph convexity for the pretrained and fine-tuned model. Error bars show the standard error of the mean, where we set $n$ to the number of data points.

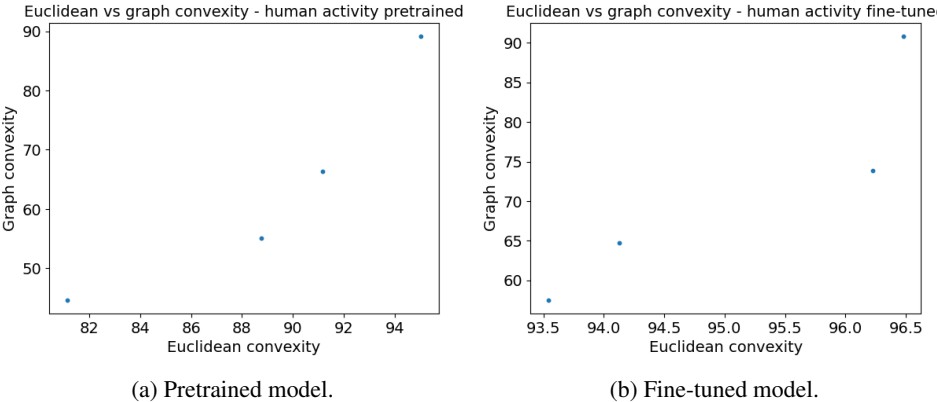

(a) Pretrained model.

(b) Fine-tuned model.

Figure 13: Human activity: Scatter plots of graph vs. Euclidean convexity for each class in the pretrained and fine-tuned model.

C.3    AUDIO DOMAIN RESULTS

Figure 14 shows the t-SNE plots of the predicted classes in the audio domain. The clustering of classes can be seen in late layers in both the pretrained and fine-tuned model, as expected it's more pronounced in the fine-tuned model.

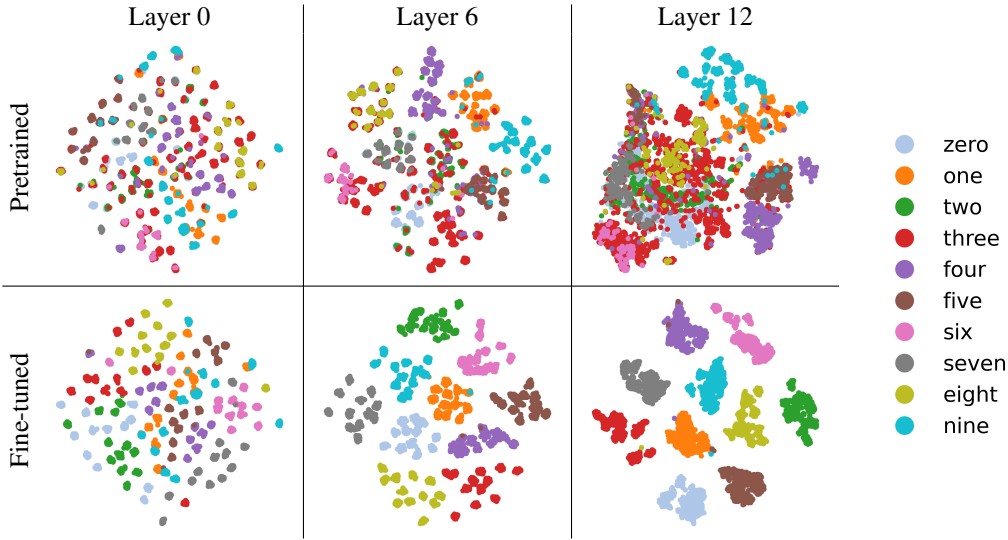

Figure 14: Audio: t-SNE plots for predicted classes, both before and after fine-tuning.

This behaviour is also observed in the convexity analysis, where the convexity generally increases for the classes throughout the layers (Figure 15). Euclidean convexity is generally higher than the graph convexity, and the convexity is higher in the fine-tuned model.

Figure 16 shows the relation between Euclidean and graph convexity per class.

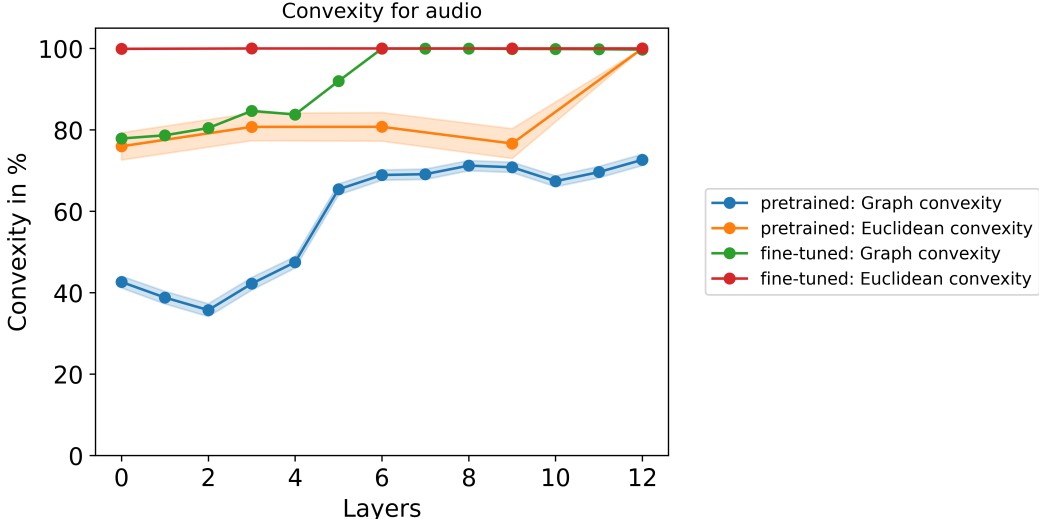

Figure 15: Audio: Euclidean and graph convexity for the pretrained and fine-tuned model. Error bars show the standard error of the mean, where we set $n$ to the number of data points. The error bars for "fine-tuned: Graph convexity" are missing.

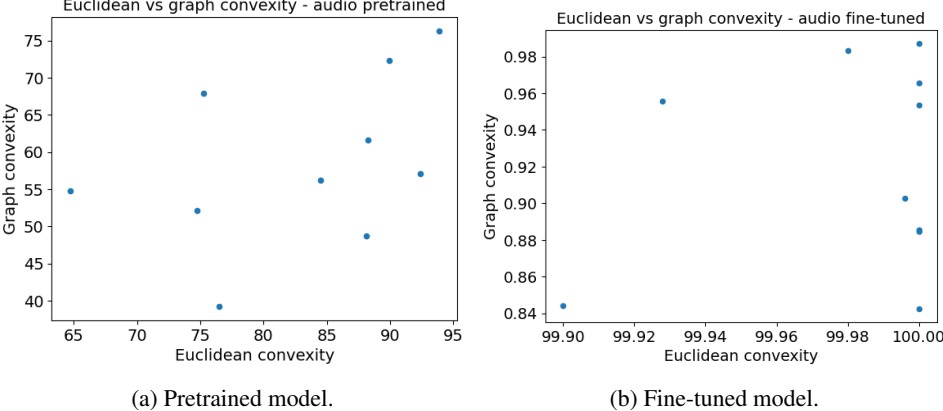

(a) Pretrained model.

(b) Fine-tuned model.

Figure 16: Audio: Scatter plots of graph vs. Euclidean convexity for each class in the pretrained and fine-tuned model.

### C.4 TEXT RESULTS

In Figure 17 we see the 2D embeddings from t-SNE for both the pretrained as well as the fine-tuned model. We see that the classes are separated through fine-tuning and that convexity increases through this process as well. Relating it to Figure 18 we can see graph convexity for the pretrained model is more or less constant throughout the network while the euclidean convexity rises sharply in the last half of the network.

The pretrained network obtained an accuracy of 60% by training a projection head from the 768-dimensional last hidden state to the 20 classes. The fully fine-tuned network obtained an accuracy of 68.6%.

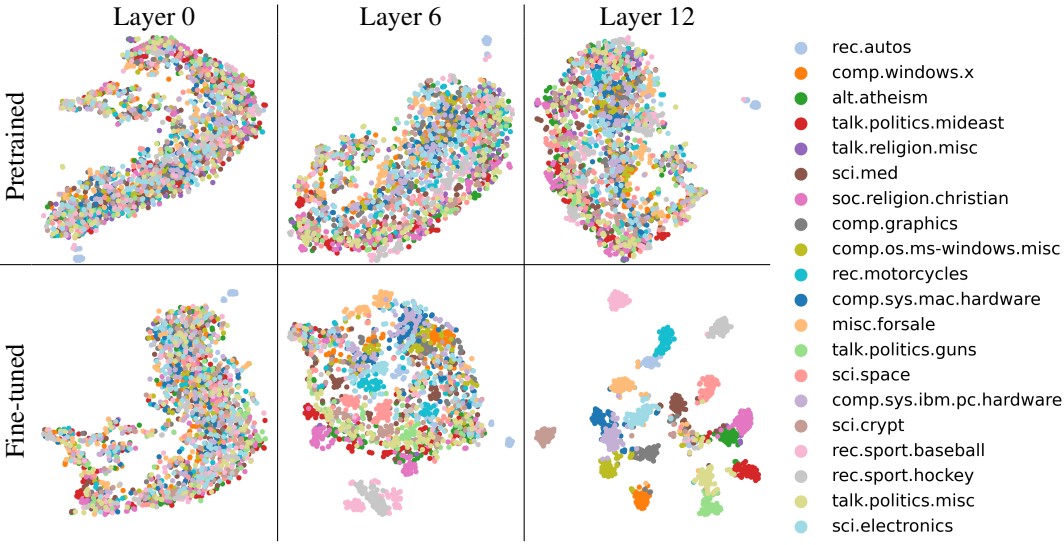

Figure 17: Text: t-SNE plots for the predicted classes both before and after fine-tuning.

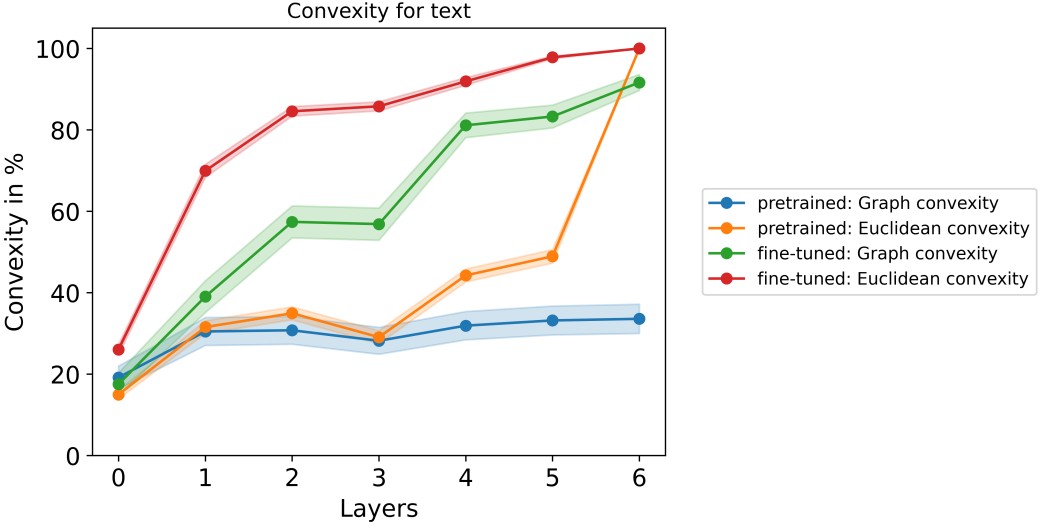

Figure 18: Text: Euclidean and graph convexity for the pretrained and fine-tuned model. Error bars show the standard error of the mean, where we set $n$ to the number of data points.

Plotting the graph and Euclidean convexity as in Figure 19 reveals that for the fine-tuned model we actually see some level of correlation between the two measurements. We also see that there is no

particular correlation between the graph convexity and euclidean convexity for the pretrained network. This can be explained by the disconnectedness of decision regions in the pretrained network which highly impacts the graph convexity.

It is also apparent from Figure 18 that the fine-tuned model does not reach 100% graph convexity in its final layer. Looking at the t-SNE embeddings in Figure 17 we see that some points with different classes are clustered closely together. This might be due to topics in the text being very closely related. This also causes the phenomena where the shortest path is traversed through the other, closely related, class, causing the graph convexity to go down.

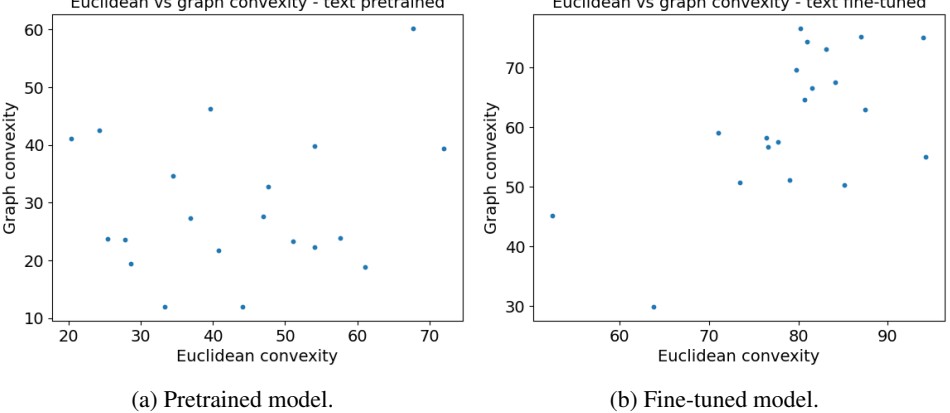

(a) Pretrained model.         (b) Fine-tuned model.

Figure 19: Text: Scatter plots of graph vs. Euclidean convexity for each class in the pretrained and fine-tuned model.

## C.5 MEDICAL IMAGING RESULTS

In Figure 20, we show t-SNE plots for the pretrained and fine-tuned model. We show embeddings after patch embedding (layer 6), after the middlemost layer in the transformer (layer 6) and just before softmax (layer 12). Clusters of different classes can already be observed in the pretrained model, however, they become more prevalent in the fine-tuned model.

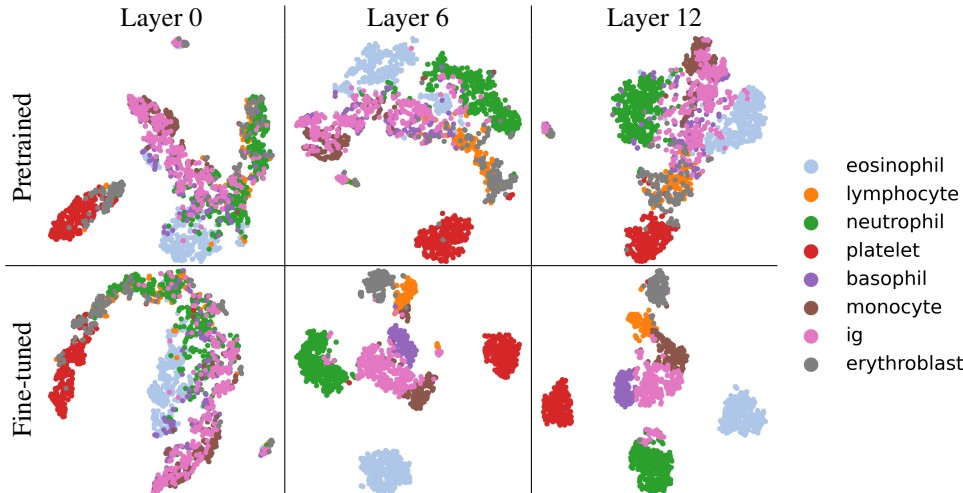

Figure 20: Medical images: t-SNE plots for the pretrainend and finetuned models.

Figure 21 shows graph and Euclidean convexity for the pretrained and fine-tuned models. Convexity increases more rapidly in the beginning for both fine-tuned and pretrained models, as well as for graph and Euclidean convexity. Graph convexity saturates at 78.4% for the pretrained model and 98.9% for the fine-tuned model. Euclidean convexity reaches 100% convexity in the final layer as expected.

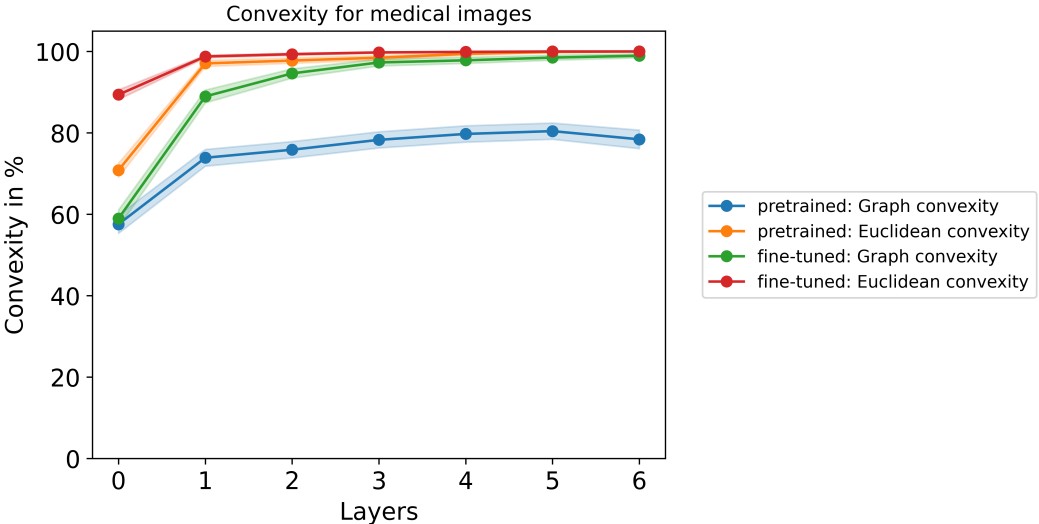

Figure 21: Medical images: Euclidean and graph convexity for the pretrained and fine-tuned model. Error bars show the standard error of the mean, where we set $n$ to the number of data points.

When comparing graph convexity with Euclidean convexity per class, on can see a positive correlation between the two convexity measures (Figure 22. Although convexity scores are generally higher in the fine-tuned model, this relation still holds.

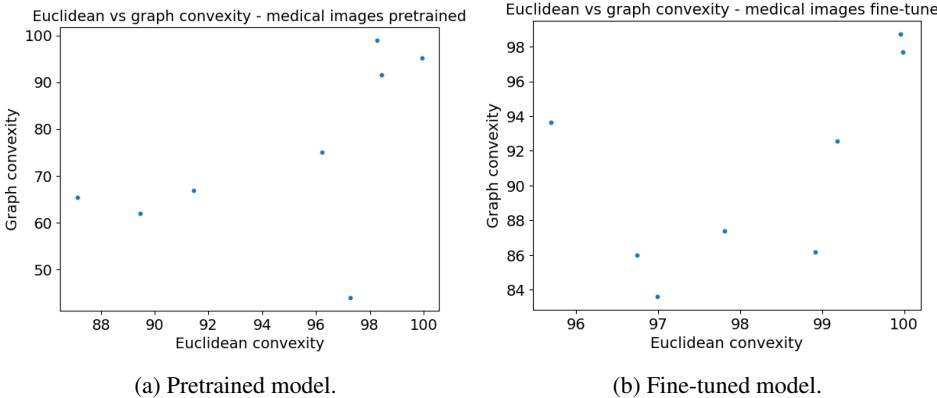

(a) Pretrained model.    (b) Fine-tuned model.

Figure 22: Medical images: Scatter plots of graph vs. Euclidean convexity for each class in the pretrained and fine-tuned model.

## C.6 K-NN VS $\epsilon$ NEIGHBORHOODS

We explore the role of the way we construct the graph. The analysis in Section 1.2.1 holds for a graph constructed with a distance cutoff $\epsilon$. We compare it to graphs constructed by keeping $K$-nearest neighbours and symmetrizing the graph. We chose the number of nearest neighbours to be 3, 5, 10 and 20. The respective $\epsilon$ values were chosen to keep approximately the same number of edges in the graph.

Figure 23b shows that the graph constructed with a distance cutoff $\epsilon$ is very disconnected, and the scores computed with this type of neighborhood are biased towards zero (see Figure 23a). If we skip the disconnected pairs and compute the score only from existing paths, the results are very close to the scores based on $K$-nearest neighbours (see Figure 23c).

Figure 23b also demonstrates that the role of the size of $K$ is negligible.

Figure 24 illustrates the difference between the data labels and the model labels in synthetic data.

## C.7 RANDOM BASELINES

We repeat the workflow described in Section 2.1 with randomly assigned labels to get a sensible meaning of the convexity scores. This is similar to measuring accuracy – the information that the accuracy of a model is $50\%$ itself does not say much about the performance since it has a completely different meaning if we have two classes (where random guessing would yield $50\%$ accuracy) or if we have 100 classes (with $1\%$ accuracy when randomly guessing). Table 1 shows the mean of baseline scores over the models for all modalities. Since we have classes of similar sizes for each modality, we see that they roughly correspond to $\frac{1}{C}$, where $C$ is the number of classes.

| Domain | Number of classes | Pretrained baseline | Fine-tuned baseline |
|---|---|---|---|
| Images | 1000 | 0.13 | 0.12 |
| Human activity | 4 | 25.05 | 25.07 |
| Audio | 10 | 9.15 | 8.51 |
| Text | 20 | 5.13 | 5.12 |
| Medical Imaging | 8 | 12.74 | 12.09 |

Table 1: Graph convexity scores in % for random baselines.

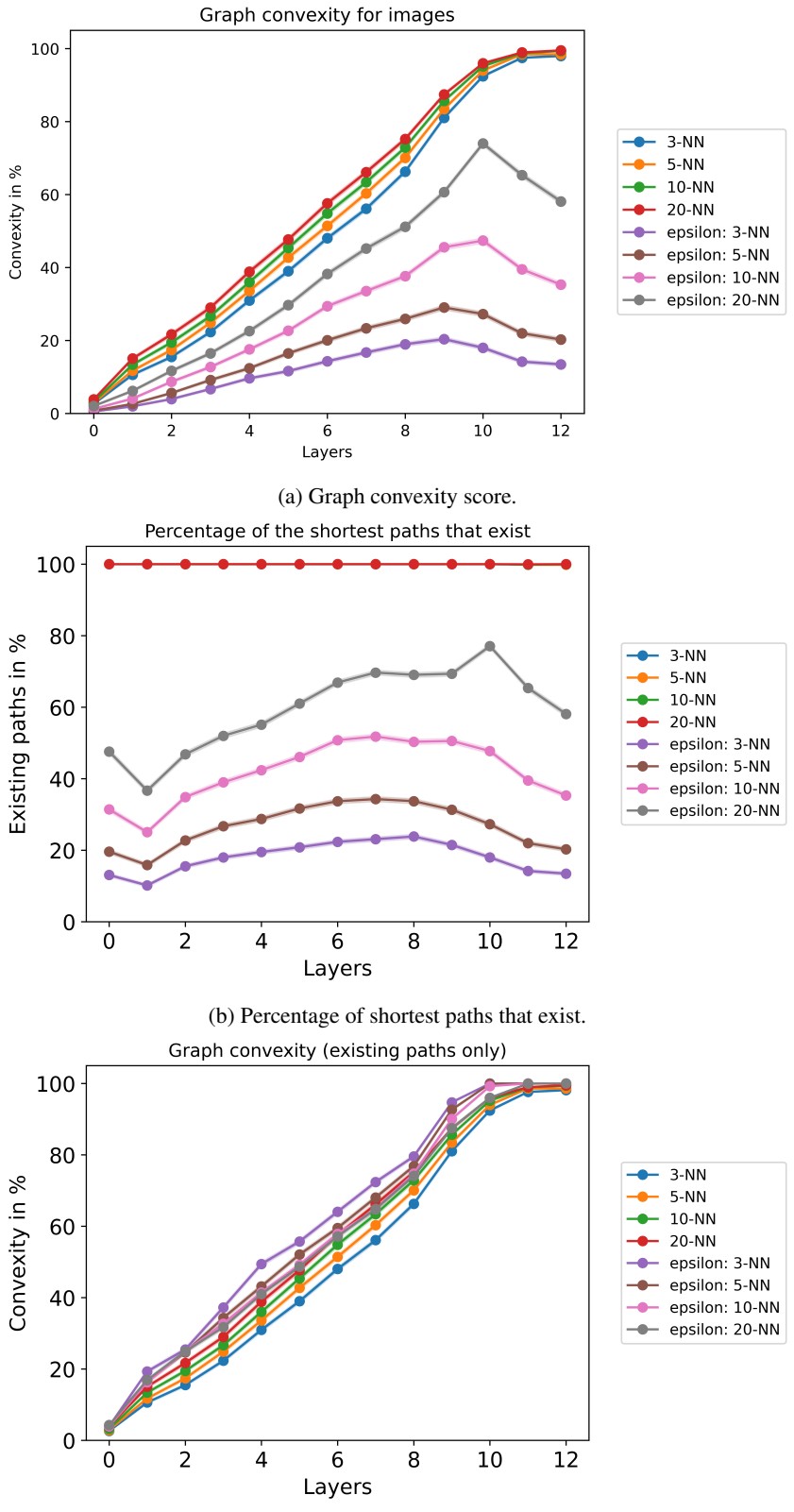

(a) Graph convexity score.

(b) Percentage of shortest paths that exist.

(c) Graph convexity score computed only if the shortest paths exist.

Figure 23: Analysis of the influence on graph convexity when using the $\epsilon$ or the K-NN approach for constructing the neighbourhood graph - fine-tuned model for images, evaluated 100 classes (5000 images).

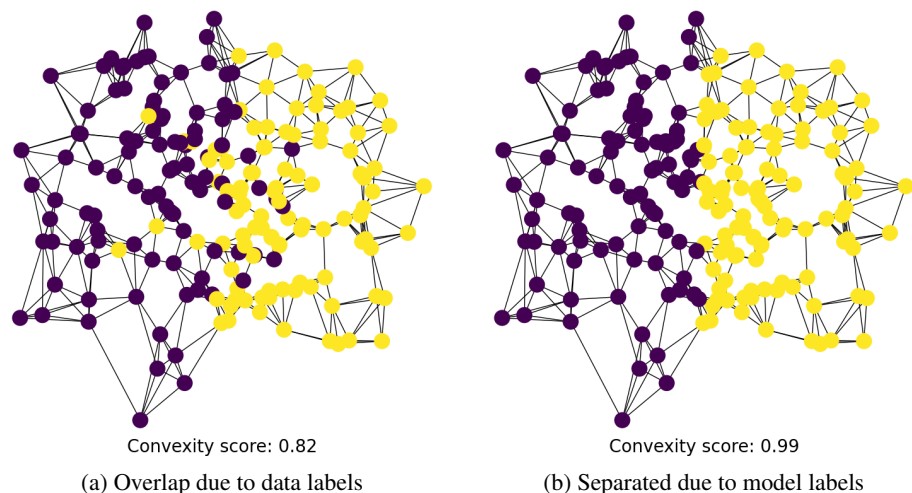

Convexity score: 0.82                    Convexity score: 0.99

(a) Overlap due to data labels            (b) Separated due to model labels

Figure 24: Difference between computing the graph convexity scores of the data labels vs. the model labels in the last layer. Model labels create Euclidean-convex regions (b), which is not the case for data labels (a) if the model's accuracy is lower than 100%.

