# OpenReview forum: "On convex decision regions in deep network representations"
_ICLR.cc/2024/Workshop/Re-Align — ICLR 2024 Workshop Re-Align Poster_

### Official Review · Reviewer_SkFz · 2024-02-23
**Review: On convex decision regions in deep network representations**

**Rating:** 3
**Fit:** 3
**Confidence:** 2

**Workshop Review:**

In the paper under review, the authors present an novel idea of quantifying the convexity of activations in specific layers of trained models. They provide a comprehensive recapitulation of the various properties of different convex spaces and introduce a methodology for measuring the convexity of latent spaces. The authors empirically demonstrate that the convexity of a class decision region in a pretrained model can predict the labelling accuracy of that class following fine-tuning.

 - *Clarity*: The paper is well-articulated, offering a broad overview of existing works. The narrative of the paper is well-structured, and the reasoning and results are clearly articulated.
 - *Correctness*: The claims within the paper are well-founded, and proofs are provided to substantiate them.
- *Novelty*: The proposed idea exhibits novelty and is interesting.
- *Interest to the Community*: This paper will be of interest to the representation alignment community due to its novel approach and relevant findings.

**Reason For Not Giving Higher Score:**

N/A

**Reason For Not Giving Lower Score:**

Positive arguments include the novelty of the idea, clear fit to the workshop theme, well-grounded claims and interesting result.

**Reviewer Domain:**

machine learning

---

### Official Review · Reviewer_WRWq · 2024-02-24
**Convexity of Deep Neural Network Decision Boundaries**

**Rating:** 3
**Fit:** 3
**Confidence:** 2

**Workshop Review:**

Summary: The authors provide a thorough review of the literature surrounding the idea of convexity in cognitive science as well as in both natural and artificial neural network representations and how this is intuitively related to generalization. They then describe precisely how this idea of convexity can be measured empirically with two different metrics (graph convexity and euclidean convexity) and provide intuitive figures to understand the tradeoffs of each. They then evaluate this measure on an extremely wide range of models covering an impressive breadth of modalities (audio, text, image, video, medical images) and demonstrate that their theoretical insights regarding convexity indeed hold up to their empirical evaluation. Finally, they demonstrate, most interestingly, that there is a significant relation between the convexity of pre-trained model decision boundaries and the resulting class-recall of these models *after fine-tuning* -- meaning that convexity appears to be predictive of model performance after fine tuning. They take this to imply that convexity is beneficial for few-shot learning.

Strengths:
- (Clarity) This paper is exceptionally well written and clear. I would argue that it is one of the clearest and most well illustrated and explained papers I have read in the past few years. I commend the authors for this feat, especially when describing such a technically complex subject.
- (Correctness) To the best of my understanding the experiments, claims, and evaluations made by the authors appear rigorous, although I did not check the proofs in detail. The main claims of the paper however do appear correct and in-line with the provided evidence.
- (Novelty) I believe measuring convexity of decision boundaries and relating it to cognitive science in this way is indeed novel and something that could be very valuable to the community.
- (Relevance) As above, this topic is timely, extremely well evaluated, and I think would be of utmost interest to the community.

Weaknesses:
- Virtually None
- Minor: On page one: "The geometrical approach is rooted in work Shepard (1987)" should use \citep.

**Reason For Not Giving Higher Score:**

N/A

**Reason For Not Giving Lower Score:**

The paper is extremely well written and evaluated. The topic is timely, interesting and novel. I see no reason to give a lower score.

**Reviewer Domain:**

machine learning

---

### Official Review · Reviewer_gZsD · 2024-02-24
**Good paper, requires generalization**

**Rating:** 2
**Fit:** 3
**Confidence:** 1

**Workshop Review:**

This work explores the concept of convexity in object regions within machine-learned latent spaces as a fundamental aspect of understanding the alignment between human and machine representations. Drawing inspiration from geometric psychology and neuroscience, the paper suggests that measuring the convexity of decision regions in learned latent spaces can be a metric for assessing model generalization ability, establishing a correlation between model convexity and fine-tuned accuracy.

**Clarity** and **Correctness**: The paper is well-written, and the experiments include various models in various domains, such as images, audio, and text. Perhaps it would be beneficial to explore generalization across other models within the same modality, given that only one model per modality has been tested.

**Novelty**: The authors investigate the convexity of decision regions in machine-learned representations for the first time, connecting the idea of convex regions in NN spaces to the hypothesis that natural concepts form convex regions in the human brain. Thus, **the paper fits the workshop topic**.

**Reason For Not Giving Higher Score:**

The paper examines neural network latent spaces, but the conclusion does not clarify whether future work can be pursued in this area. Can this analysis be leveraged to enhance neural networks, such as in training processes? Additionally, they do not provide examples of non-convex regions, and they only cite an existing work: "The resulting decision regions are, therefore, unions of convex sets and may, in general, be non-convex or non-connected, as noted in Bishop et al. (1995)." Thus, isn't the convexity of NN latent spaces trivial?

**Reason For Not Giving Lower Score:**

The paper is well suited for the workshop since it uses, for the first time, convexity as a tool to understand if there is an alignment between human and machine representations. Furthermore, the authors conducted multiple experiments to support their conclusions.

**Reviewer Domain:**

machine learning

---

### Decision · Program_Chairs · 2024-03-02

Accept (Poster)